# Learning Generalizable Shape Completion with SIM(3) Equivariance

Yuqing Wang[1*]     Zhaiyu Chen[1,2*]     Xiao Xiang Zhu[1,2]

[1]Technical University of Munich     [2]Munich Center for Machine Learning

## Abstract

3D shape completion methods typically assume scans are pre-aligned to a canonical frame. This leaks pose and scale cues that networks may exploit to memorize absolute positions rather than inferring intrinsic geometry. When such alignment is absent in real data, performance collapses. We argue that robust generalization demands architectural equivariance to the similarity group, $SIM(3)$, so the model remains agnostic to pose and scale. Following this principle, we introduce the first $SIM(3)$-equivariant shape completion network, whose modular layers successively canonicalize features, reason over similarity-invariant geometry, and restore the original frame. Under a de-biased evaluation protocol that removes the hidden cues, our model outperforms both equivariant and augmentation baselines on the PCN benchmark. It also sets new cross-domain records on real driving and indoor scans, lowering minimal matching distance on KITTI by $17\%$ and Chamfer distance $\ell 1$ on OmniObject3D by $14\%$. Perhaps surprisingly, ours under the stricter protocol still outperforms competitors under their biased settings. These results establish full $SIM(3)$ equivariance as an effective route to truly generalizable shape completion. Project page: https://sime-completion.github.io.

## 1 Introduction

3D scans are often riddled with gaps due to occlusions and limited sensor coverage. Completing the missing geometry lets robots plan stable grasps, autonomous vehicles reason about hidden traffic, and curators digitize heritage artifacts without repeated scanning [1, 2, 3, 4]. However, most shape completion methods [5, 6, 7, 8] are developed on curated benchmarks where every scan is pre-aligned to a canonical frame with a fixed pose and scale relative to ground truth. These leaked cues inadvertently bias learning: instead of inferring intrinsic geometry, neural networks tend to memorize where shapes reside in that frame, leading to inflated performance that collapses once the alignment is removed in practice [9, 10]. The resulting gap between benchmark success and real-world reliability highlights the challenge of exploiting geometry without inheriting extrinsic transforms that convey it.

Data augmentation mitigates this alignment bias by randomizing transforms during training to approximate inference-time invariance, but it entangles those transforms with underlying geometry and leaves the core ambiguity unresolved. Architectural equivariance, by contrast, aims to ensure that applying a transform to the input induces the same transform in the prediction, thereby isolating geometry from transforms and sharpening learned representations [11, 12]. However, existing equivariant methods still struggle to enforce this separation. $SO(3)$-equivariant shape completion [13, 14] typically normalizes inputs using ground-truth centroids and scales, while $SE(3)$-equivariant variants [10, 15, 16] still rely on ground-truth scale to canonicalize scans. Relying on such privileged information effectively reduces these models to explicit canonicalization (Fig. 1), undermining the true purpose of equivariance. To our knowledge, no existing architecture fully eliminates alignment bias, as all still require some ground-truth alignment that is unavailable in practice.

---

[*] Equal contribution. Corresponding author: zhaiyu.chen@tum.de.

39th Conference on Neural Information Processing Systems (NeurIPS 2025).

We argue that true generalization hinges on handling arbitrary similarity transforms, SIM(3), including rotation, translation, and scaling. To this end, we present the first shape completion architecture whose modules are SIM(3)-equivariant by design. During training, the network learns representations agnostic to pose and scale; at test time, any similarity transform applied to the input induces identical changes in the prediction (Fig. 1). To recover the completed shape in the sensor frame, we introduce a lightweight restoration path that re-injects the transform information progressively. By disentangling

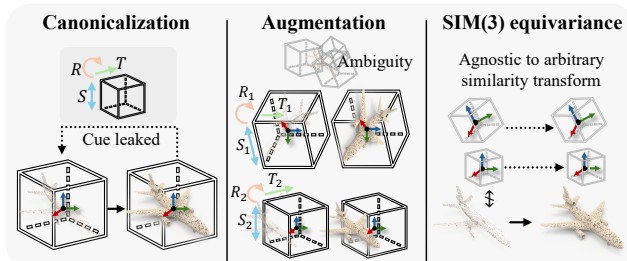

Figure 1: **Three paradigms for shape completion.** Explicit canonicalization, including SO(3)- and SE(3)-equivariant variants, leak pose and scale cues and fail on non-canonical inputs. Data augmentation mitigates the alignment bias but incurs ambiguity. We present a SIM(3)-equivariant approach that generalizes to arbitrary similarity transforms.

intrinsic geometry from extrinsic transforms, our model trained on synthetic data transfers directly to real scans under a fair, de-biased evaluation protocol. In summary, our contributions include:

1. **Problem identification.** We reveal pose and scale bias in existing shape completion methods, and identify SIM(3) equivariance as a prerequisite for reliable, in-the-wild generalization.

2. **Generalizable framework.** We develop the *first* fully SIM(3)-equivariant network for shape completion. It integrates feature canonicalization, similarity-invariant geometric reasoning, and a transform restoration path into a modular design, generalizing from synthetic to real scans.

3. **Protocol and resources.** We establish a rigorous evaluation protocol that eliminates hidden pose and scale bias, release code for reproducibility, and provide thorough analyses that pinpoint where equivariance delivers its gains. Under this protocol our method sets a new state of the art.

## 2 Related Work

**Equivariant 3D representations.** Equivariant neural networks learn features that transform consistently under input symmetry operations. Grounded in group theory and representation learning [11, 17, 18, 19, 20, 21, 22, 23, 24, 25, 26, 27], 3D equivariant models have been developed to handle variability in data transforms. One approach employs group convolution [28, 29, 30] to encode symmetry, but these methods often remain bound to specific architectures and lack generality. Another leverages tensor algebra with spherical harmonics as irreducible representations to achieve equivariance [12, 31]. Vector neurons (VN) [32] replaced high-order tensors with structured 3D vectors, providing a modular SO(3)-equivariant alternative. Subsequent extensions integrated attention mechanisms [33] and translation equivariance [34], yet most VN-based networks remain relatively shallow, which limits their applicability to complex 3D tasks. Nonetheless, both paradigms have driven advances in 6-DoF pose estimation [35, 36, 37], point cloud registration [38, 39], robotic manipulation [40, 41], and 3D reconstruction [42, 43, 44], among others [45, 46]. However, most methods are confined to SO(3) or SE(3) equivariance and require centering or scale normalization. Both assumptions break down in real-world scenarios without ground truths. Although SIM(3)-equivariance can overcome these limitations, existing efforts [40, 41, 46] remain sparse, depend on near-complete inputs, and struggle on partial observations. We close this gap with a fully SIM(3)-equivariant Transformer architecture.

**3D shape completion.** Early methods represented geometry in voxels and applied 3D CNNs [5, 47, 48, 49], but cubic complexity limited resolution. The use of symmetric functions for permutation invariance [50] led to the development of shape completion networks that directly consume 3D points [51, 52]. More recently, Transformers [53] have recast shape completion as set-to-set translation [6], and now lead the field [7, 8, 54, 55, 56, 57]. Almost all prior work, however, presumes that inputs are pre-aligned to the training frame, letting pose and scale cues leak into models and collapse performance on raw scans without special adaptation [9, 58]. Existing remedies follow two paradigms. Data augmentation randomizes transforms during training, but entangles extrinsic transforms with intrinsic geometry and incurs ambiguity at test time. Equivariance-based methods

either estimate a canonical pose prior to completion [15, 16] or replace standard layers with equivariant variants [10, 13]. The former relies on a fragile pose estimator that misaligns under partial observations and propagates errors for the downstream completion. The latter often loses fine-grained details and underperforms Transformer models with data augmentation [13]. A recent anchor-point scheme extends equivariance to SE(3) [10], but its dependence on brittle anchor selection hampers performance and still falls behind augmented baselines. Moreover, all these methods still rely on ground-truth bounding boxes to cancel scale variance, re-introducing the very cues they aim to discard. In contrast, we integrate full SIM(3) equivariance into every layer, inherently agnostic to arbitrary pose and scale, delivering the first shape completion method that truly generalizes to completely unaligned real-world scans.

## 3 Method

### 3.1 Preliminaries

**Formulation.** Shape completion, $f_\theta \colon \mathbf{x} \to \hat{\mathbf{y}}$, takes a partial observation $\mathbf{x} = \{x_i \in \mathbb{R}^3\}_{i=1}^{N_{\text{in}}}$ (*e.g.*, a point set) and aims to reconstruct a set $\hat{\mathbf{y}} = \{\hat{y}_i \in \mathbb{R}^3\}_{i=1}^{N_{\text{out}}}$ representing the completed shape, both expressed in the original sensor frame. Due to varying capture conditions, $\mathbf{x}$ and its ground truth $\mathbf{y}$ may undergo a shared unknown similarity transform $g = (s, R, t) \in \text{SIM}(3)$:

$$\mathbf{x}' := g \cdot \mathbf{x} = sR\mathbf{x} + t, \qquad \mathbf{y}' := g \cdot \mathbf{y} = sR\mathbf{y} + t, \qquad s \in \mathbb{R}_+, \ R \in \text{SO}(3), \ t \in \mathbb{R}^3, \quad (1)$$

where $\mathbf{x}'$ and $\mathbf{y}'$ are transformed representations of the same object. Although the transform $g$ alters coordinates, the intrinsic geometry remains invariant. To guarantee consistent predictions under any similarity transform, we enforce SIM(3) equivariance in $f_\theta$. Paired with a permutation-invariant loss $\mathcal{L}\big(f_\theta(\mathbf{x}), \mathbf{y}\big)$ (*e.g.*, Chamfer distance), we solve the constrained optimization problem:

$$\min_\theta \ \mathcal{L}\big(f_\theta(\mathbf{x}), \mathbf{y}\big) \quad \text{s.t.} \quad f_\theta(g \cdot \mathbf{x}) = g \cdot f_\theta(\mathbf{x}) \ \ \forall g \in \text{SIM}(3). \tag{2}$$

**Vector neurons.** To enforce the equivariance constraint in Eq. (2), we build on the vector neuron (VN) framework [32], which replaces scalar neurons with 3D vector ones. At layer $l$, we organize $D^l$ vector channels into $M$ vector features:

$$\mathcal{V}^l = \{\mathbf{V}_i^l\}_{i=1}^M, \quad \mathbf{V}_i^l \in \mathbb{R}^{D^l \times 3}. \tag{3}$$

The VN framework defines linear, nonlinear, and pooling operations on these vector neurons to preserve symmetry under the SO(3) group action. We adapt VN representations and integrate these operations as building blocks into our SIM(3)-equivariant architecture. For simplicity, we omit the layer index $l$ in the following unless cross-layer operations are involved. See Appendix C for details.

**Challenges.** We decompose the overall objective in Eq. (2) into three complementary requirements, separating geometric reasoning from transform alignment:

$$\min_\theta \begin{cases} \mathcal{L}\big(g^\star \cdot f_\theta(\mathbf{x}), \ \mathbf{y}\big) & \textbf{Req. (1)} \ \text{geometric reasoning} \\ \|g^\star - \mathrm{I}\| & \textbf{Req. (3)} \ \text{transform alignment} \end{cases} \quad \text{s.t.} \quad \underbrace{f_\theta(g \cdot \mathbf{x}) = g \cdot f_\theta(\mathbf{x})}_{\textbf{Req. (2)} \ \text{equivariance}}. \tag{4}$$

Here $g^\star$ denotes the optimal alignment between the prediction and the ground truth, such that $\mathcal{L}\big(g^\star \cdot f_\theta(\mathbf{x}), \ \mathbf{y}\big)$ measures geometric discrepancy independent of pose and scale. In this formulation:

**Req. (1) Geometric reasoning.** The network must infer the complete geometry of missing regions, even when $\mathbf{x}$ are sparsely and heterogeneously sampled. This demands strong structural priors and fine-grained feature extraction that shallow architectures cannot achieve.

**Req. (2) Equivariance.** The model must respect SIM(3) symmetry, which needs to be enforced throughout, because any single layer that is not equivariant will break global equivariance. Thus, each operator must be redesigned to commute with the group actions.

**Req. (3) Transform alignment.** The completed shape $f_\theta(\mathbf{x})$ must be presented in the sensor frame so that no further alignment is required for downstream tasks. Equivalently, $g^\star$ should remain close to the identity transform I under any reasonable metric $\|\cdot\|$. This requires propagating pose and scale information throughout the network to preserve the original frame.

The next section details how each component of our architecture satisfies these requirements.

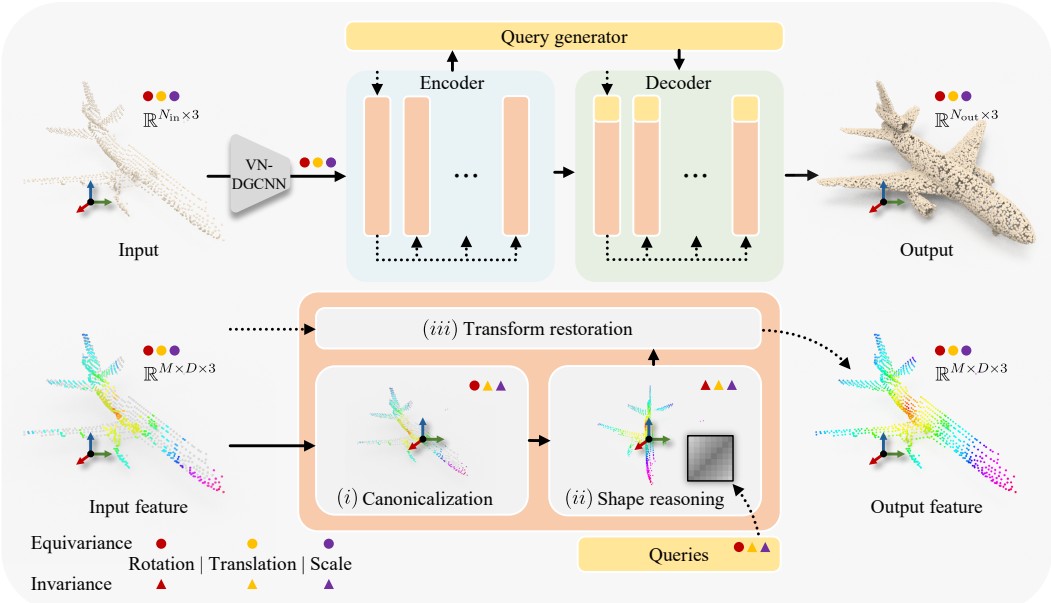

Figure 2: **Overview of our** $\text{SIM}(3)$**-equivariant shape completion pipeline.** We extract point patch features with VN-DGCNN [32] and feed them into a Transformer encoder-decoder. Within each layer module, we $(i)$ canonicalize features to be translation- and scale-invariant, $(ii)$ reason intrinsic geometry via $\text{SIM}(3)$-invariant attention, and $(iii)$ restore the original transform. This guarantees that both intermediate features and the reconstructed shape adhere to $\text{SIM}(3)$ transforms.

## 3.2 $\text{SIM}(3)$-equivariant shape completion

We progressively address the three challenges in Eq. (4) via $L$ $\text{SIM}(3)$-equivariant blocks, which enforce the equivariance constraint as in Req. (2) by design:

$$\mathcal{B}^l = \mathcal{R}^l \circ \mathcal{A}^l \circ \mathcal{C}^l, \quad f_\theta(x) = \mathcal{B}^L \circ \cdots \circ \mathcal{B}^1(x). \tag{5}$$

Each block $\mathcal{B}^l$ comprises three sequential stages (Fig. 2): $(i)$ feature canonicalization $\mathcal{C}^l$ produces translation- and scale-invariant feature vectors; $(ii)$ similarity-invariant shape reasoning $\mathcal{A}^l$ optimizes Req. (1); and $(iii)$ pose and scale are restored via $\mathcal{R}^l$ to satisfy the transform-alignment objective in Req. (3). By stacking these blocks and end-to-end optimizing Eq. (4), the network iteratively refines its prediction and converges to the completed shape expressed in the input frame.

**Canonicalization.** Robust geometric reasoning requires removing transform variance embedded in the feature representation. As shown in Fig. 3, we extend layer normalization [33, 59, 60] to explicitly factor out global translation and scale from shape features. Specifically, let $\bar{\mathbf{V}}_i \in \mathbb{R}^3$ denote the channel-wise mean of the latent vectors. We first subtract this mean from each $\mathbf{V}_i$ to eliminate translation, then normalize the centered vector to remove scale variation, followed by a vanilla layer normalization applied to the row-wise norm $\left\| \mathbf{V}_i - \bar{\mathbf{V}}_i \right\|_2 \in \mathbb{R}^{D \times 1}$ to stabilize training without altering the direction of the vector neurons, thus preserving rotational consistency. Formally, the ex-

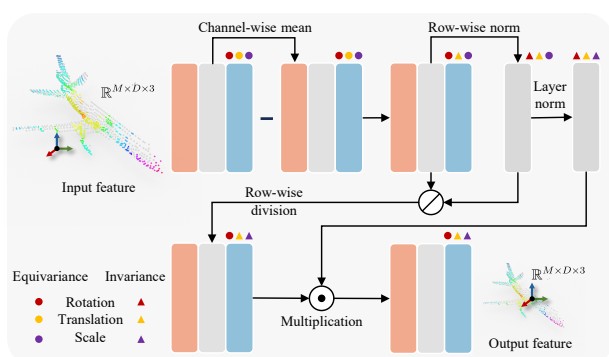

Figure 3: **Layer normalization.** We extend standard layer normalization [59] to producing translation- and scale-invariant features and preserving rotation equivariance.

tended layer normalization is defined as:

$$\mathcal{C}^l : \quad \mathbf{V}'_i = \mathrm{layernorm}\big(\big\|\mathbf{V}_i - \bar{\mathbf{V}}_i\big\|_2\big)\frac{\mathbf{V}_i - \bar{\mathbf{V}}_i}{\big\|\mathbf{V}_i - \bar{\mathbf{V}}_i\big\|_2}. \tag{6}$$

This procedure canonicalizes features to an implicitly defined canonical feature frame.

**Shape reasoning.** In the scale- and translation-invariant canonical feature frame after $\mathcal{C}^l$, we perform similarity-invariant shape reasoning via the rotation-invariant attention weights $\mathbf{A}$ from VN-Transformer [33], ensuring no residual rotational bias. Since attention is invariant under any $g \in \mathrm{SIM}(3)$, the local form of the objective in Req. (1), for each shape reasoning layer, reduces to:

$$\mathcal{A}^l : \quad \min_{\theta_A}\ \mathcal{L}\big(f_{\theta_A}(\mathbf{A}(\mathbf{x})), \mathbf{y}\big), \tag{7}$$

$$\text{where} \quad a_{ij} = \mathrm{softmax}_j\left(\tfrac{1}{\sqrt{3D}}\ \langle \mathbf{W}_Q\mathbf{V}'_i, \mathbf{W}_K\mathbf{V}'_j\rangle_F\right), \quad a_{ij} \in \mathbf{A}(\mathbf{x}). \tag{8}$$

Here, the query and key projections $\mathbf{W}_Q, \mathbf{W}_K \in \mathbb{R}^{D \times D}$, included in the model parameters $\theta$, define how shape features interact. The Frobenius inner product $\langle\cdot\rangle_F$ is invariant to joint rotation of $\mathbf{V}'_i$ and $\mathbf{V}'_j$, making the attention weights depend solely on their relative geometry. The attention weights satisfy $\mathbf{A}(g\cdot\mathbf{x}) = \mathbf{A}(\mathbf{x})$ for any $g \in \mathrm{SIM}(3)$, which decouples intrinsic shape features from transforms.

**Transform restoration.** $\mathrm{SIM}(3)$ equivariance alone does not guarantee that the shape reasoning output is aligned with the original sensor frame, as it preserves only relative pose and scale. The final challenge is to recover this absolute alignment, as required in Req. (3). To achieve this, we introduce a transform restoration path to propagate input pose and scale via residual connections (Fig. 2). After each $\mathrm{SIM}(3)$-invariant shape reasoning step, the restoration path reinjects translation and scale to recover spatial grounding. Rotation is implicitly preserved through the attention output $\mathbf{Z}_i = \sum_j a_{ij}\mathbf{W_V}\mathbf{V}'_j$, where $\mathbf{W_V}$ is the value projection weight. Translation and scale are injected in accordance with their group actions via addition and multiplication, respectively:

$$\mathcal{R}^l : \quad \mathbf{V}^{l+1} = \mathbf{V}^l + \Phi(\mu^l \mathbf{Z}), \tag{9}$$

where $\mu^l = \mathbb{E}_{D^l}\big\|\mathbb{E}_i(\mathbf{V}_i^l - \bar{\mathbf{V}}_i^l)\big\|_2$ is a global scale statistic computed from the average norm of centered input features, and $\Phi$ is a VN linear layer that fuses spatial and geometric features to guide alignment. By restoring translation and scale at each stage, we reestablish full $\mathrm{SIM}(3)$ equivariance at the module output (see Appendix C for the proof), ensuring consistent spatial grounding across layers, as shown in Fig. 4.

### 3.3 Network architecture

We build on the AdaPoinTr [7] backbone, which features a coarse-to-fine shape completion scheme. We replace the original DGCNN with VN-DGCNN [32] for local geometric feature extraction while retaining $\mathrm{SIM}(3)$ equivariance, and replace every Transformer layer with our $\mathrm{SIM}(3)$-equivariant module introduced in Sec. 3.2. Key components, such as the query generator and the reconstruction head, are likewise implemented to be equivariant or invariant when appropriate. The network takes a partial input point cloud with 2,048 ($N_{\mathrm{in}}$) points and predicts a complete shape of 16,384 ($N_{\mathrm{out}}$) points. Aside from these changes, we retain AdaPoinTr's network depth, loss function, and training settings to

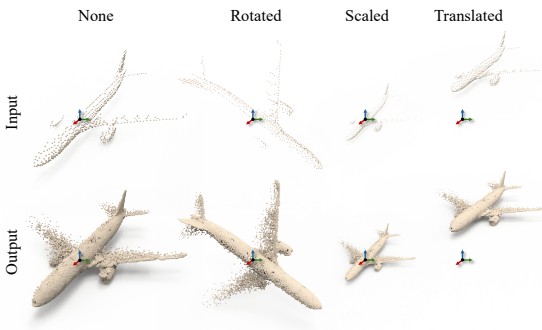

Figure 4: $\mathrm{SIM}(3)$ **equivariance.** Our outputs follow arbitrary similarity transforms applied to inputs.

ensure a fair comparison. Fig. 2 illustrates the architecture. Further architecture and implementation details are provided in Appendix C and D.

# 4 Experiments

## 4.1 Experimental setup

**Datasets and baselines.** We first evaluate on the PCN benchmark [51], which comprises eight categories from ShapeNet [61] with paired partial and complete point clouds. To assess cross-domain transferability, we directly apply PCN-trained models, without further normalization, to real-world scans from KITTI [62] and OmniObject3D [63]. We compare against leading non-equivariant shape completion methods, namely PoinTr [6], SeedFormer [54], SnowflakeNet [64], AnchorFormer [8], and AdaPoinTr [7], each trained with $\mathrm{SIM}(3)$ augmentations for fairness. Because no prior model offers full $\mathrm{SIM}(3)$ equivariance, we resort to including the $\mathrm{SO}(3)$-equivariant EquivPCN [13] and the $\mathrm{SE}(3)$-equivariant SCARP [15] and ESCAPE [10] as baselines.

**Evaluation protocol.** For our model, which requires no training-time augmentation, we adopt the train/test setting of $\mathrm{I}/\mathrm{SIM}(3)$ where I denotes the identity transform. Each baseline is first evaluated under the group it was designed for (*i.e.*, $\mathrm{I}/\mathrm{SO}(3)$ for EquivPCN [13], $\mathrm{I}/\mathrm{SE}(3)$ for ESCAPE [10], and $\mathrm{SE}(3)/\mathrm{SE}(3)$ for SCARP [15]) to reveal their upper-bound performance when pose/scale cues are still partly available. We then report their performance under $\mathrm{SIM}(3)/\mathrm{SIM}(3)$ with additional data augmentation. On PCN, rotations are sampled uniformly from $\mathrm{SO}(3)$, while each partial input is *itself* centered and scaled to the unit sphere. This prevents cue leakage from ground truths and realistically simulates real-world inputs that models actually have access to. On KITTI and OmniObject3D, we transfer our model without any cue leakage, while competing equivariant methods receive canonicalized inputs via ground-truth alignment; otherwise, they fail completely. For PCN and OmniObject3D, we report Chamfer distance $\ell_1$ (CD-$\ell_1$, scaled by $10^3$) and F-score@1% (F1). For experiments on KITTI, we follow prior practices [6, 7, 8, 54] and report the Fidelity and Minimal Matching Distance (MMD) metrics. All metrics are computed in the common canonical frame with unit scale for direct comparison. We refer to our method as **SIMECO** in all comparisons.

## 4.2 De-biased benchmark evaluation

**Against data augmentation.** Table 1 compares our $\mathrm{SIM}(3)$-equivariant model against leading non-equivariant networks trained with augmentation. Our method achieves the *lowest* average CD-$\ell_1$ and the *highest* F1 score, outperforming AdaPoinTr by 10% and 8%, respectively. It yields the best score in every category, with consistent error reductions across the board. Qualitative results in Fig. 5 show that our completions faithfully recover fine geometric details such as sharp airplane wings, slender lamp stems, and thin table legs, whereas the augmentation-based baseline produces blurrier or distorted shapes. Notably, AdaPoinTr without augmentation collapses under the de-biased protocol. These results confirm the superiority of our architectural equivariance over heavy data augmentation.

Table 1: **Evaluation on PCN.** We compare methods supporting only $\mathrm{SO}(3)$ (top) and $\mathrm{SE}(3)$ (middle), and those with $\mathrm{SIM}(3)$ augmentation (bottom). "Transform" indicate train/test settings. Our model outperforms competitors limited to partial transform groups and those with data augmentation. CD-$\ell_1$ values are scaled by a factor of 1000. **Bold** numbers indicate the best $\mathrm{SIM}(3)$ results.

| Method | Transform | Airpl. | Cab. | Car | Chair | Lamp | Sofa | Table | Wat. | CD-$\ell_1$ ↓ | F1 ↑ |
|---|---|---|---|---|---|---|---|---|---|---|---|
| EquivPCN [13] | $\mathrm{I}/\mathrm{SO}(3)$ | 8.38 | 13.74 | 11.81 | 14.31 | 12.50 | 15.68 | 12.86 | 11.02 | 12.54 | 0.569 |
| AdaPoinTr [7] | $\mathrm{SO}(3)/\mathrm{SO}(3)$ | 5.76 | 11.27 | 9.63 | 9.61 | 6.71 | 11.53 | 8.15 | 7.81 | 8.81 | 0.693 |
| SCARP [15] | $\mathrm{SE}(3)/\mathrm{SE}(3)$ | 10.05 | 40.82 | 23.02 | 22.92 | 29.17 | 62.51 | 57.82 | 37.59 | 35.49 | 0.223 |
| ESCAPE [10] | $\mathrm{I}/\mathrm{SE}(3)$ | 8.13 | 13.18 | 10.43 | 10.62 | 8.07 | 13.74 | 9.35 | 9.81 | 10.41 | 0.650 |
| AdaPoinTr [7] | $\mathrm{SE}(3)/\mathrm{SE}(3)$ | 6.16 | 12.56 | 10.48 | 9.77 | 7.10 | 12.36 | 8.34 | 8.57 | 9.42 | 0.685 |
| *Evaluation under de-biased protocol* | | | | | | | | | | | |
| AdaPoinTr [7] | $\mathrm{I}/\mathrm{SIM}(3)$ | 31.93 | 78.90 | 63.51 | 60.56 | 61.47 | 70.77 | 71.94 | 42.74 | 60.23 | 0.206 |
| EquivPCN [13] | $\mathrm{SIM}(3)/\mathrm{SIM}(3)$ | 9.10 | 14.60 | 13.09 | 15.74 | 13.74 | 16.42 | 14.74 | 11.74 | 13.65 | 0.523 |
| ESCAPE [10] | $\mathrm{SIM}(3)/\mathrm{SIM}(3)$ | 12.59 | 22.54 | 18.63 | 15.86 | 12.68 | 24.38 | 14.17 | 14.48 | 16.88 | 0.515 |
| PoinTr [6] | $\mathrm{SIM}(3)/\mathrm{SIM}(3)$ | 10.18 | 17.97 | 15.61 | 16.94 | 13.39 | 16.80 | 17.75 | 12.18 | 15.10 | 0.434 |
| SeedFormer [54] | $\mathrm{SIM}(3)/\mathrm{SIM}(3)$ | 8.42 | 17.32 | 15.08 | 12.10 | 8.25 | 17.19 | 11.38 | 9.62 | 12.42 | 0.616 |
| Snowflake [64] | $\mathrm{SIM}(3)/\mathrm{SIM}(3)$ | 7.99 | 15.59 | 13.81 | 11.89 | 8.58 | 15.66 | 10.59 | 9.72 | 11.73 | 0.621 |
| AnchorFormer [8] | $\mathrm{SIM}(3)/\mathrm{SIM}(3)$ | 7.77 | 13.61 | 12.13 | 12.71 | 9.16 | 14.26 | 10.95 | 9.35 | 11.24 | 0.599 |
| ODGNet [56] | $\mathrm{SIM}(3)/\mathrm{SIM}(3)$ | 6.16 | 11.60 | 11.15 | 10.13 | 6.81 | 13.12 | 9.48 | 8.19 | 9.58 | 0.659 |
| AdaPoinTr [7] | $\mathrm{SIM}(3)/\mathrm{SIM}(3)$ | 6.46 | 12.17 | 10.51 | 10.29 | 7.59 | 12.26 | 8.90 | 8.14 | 9.54 | 0.661 |
| **SIMECO** (ours) | $\mathrm{I}/\mathrm{SIM}(3)$ | **6.02** | **10.75** | **9.27** | **9.25** | **6.66** | **11.16** | **7.82** | **7.77** | **8.59** | **0.714** |

**Against equivariant networks.** Table 1 benchmarks our method against other SO(3)- and SE(3)-equivariant networks, each evaluated under its native transform group. In contrast, by tackling the full SIM(3) group, we address a substantially harder setting. Despite this, our model reduces average CD-$\ell_1$ from 10.41 to 8.59 ($-17\%$) and raises F1 from 0.650 to 0.714 ($+10\%$) relative to ESCAPE, which uses the same AdaPoinTr backbone. EquivPCN (SO(3)) and SCARP (SE(3)) lag even further, confirming that full SIM(3) equivariance enables learning more intrinsic shape representations. Moreover, neither EquivPCN nor ESCAPE can outperform augmentation-based baselines in their respective groups. And training ESCAPE and EquivPCN with SIM(3) augmentation degrades their performance, highlighting that equivariance not built into the architecture is hard to acquire through augmentation alone. Figs 5 and 6 respectively demonstrate that our model preserves fine details and delivers consistent outputs under various pose and scale perturbations.

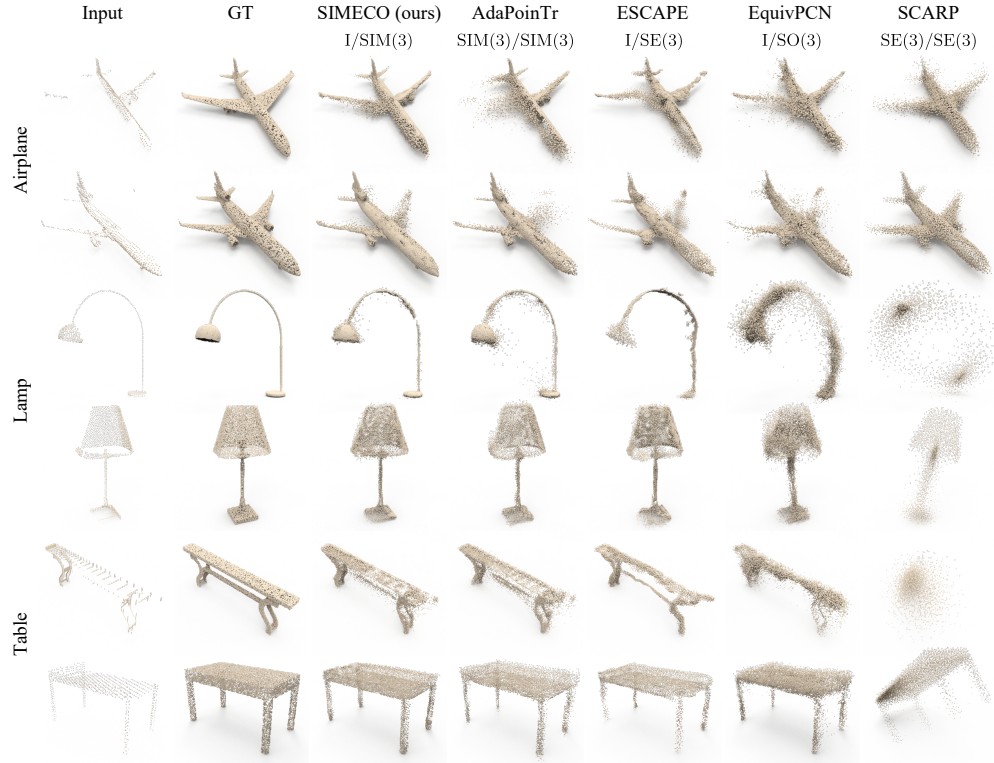

Figure 5: **Comparison on PCN.** Our SIM(3)-equivariant model outperforms other equivariant methods restricted to SO(3) and SE(3) and non-equivariant baseline trained with SIM(3) augmentation.

### 4.3 Cross-domain generalization

**Unseen driving scans (KITTI).** Table 2 presents cross-domain performance on KITTI using models trained solely on synthetic, canonicalized PCN data. Even under full SIM(3) variation, our model reduces MMD from 6.47 to 5.35 ($-17\%$) compared to the strongest non-equivariant baseline and cuts ESCAPE's Fidelity error from 1.81 to 0.56 ($-69\%$). EquivPCN achieves even lower MMD, but only under its native SO(3) setting. Crucially, all competing methods, apart from those using data augmentation, rely on ground-truth bounding boxes to normalize KITTI inputs. SO(3) models use them for translation and scale normalization; SE(3) models use them for scale. This requirement leaks information and is impractical in real deployments. By contrast, our fully SIM(3)-equivariant architecture requires *no* external normalization. In Fig. 7, our model recovers car wheels and indoor details more faithfully, while AdaPoinTr with augmentation produces oversmoothed outputs.

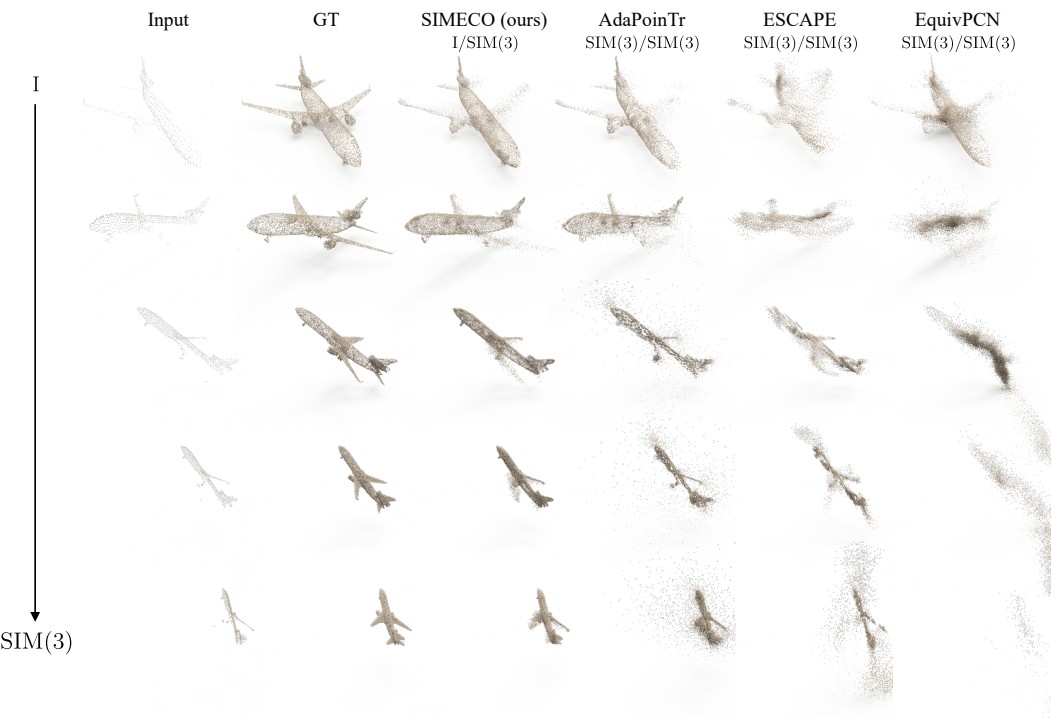

|  | Input | GT | SIMECO (ours) I/SIM(3) | AdaPoinTr SIM(3)/SIM(3) | ESCAPE SIM(3)/SIM(3) | EquivPCN SIM(3)/SIM(3) |

Figure 6: **Robustness to pose and scale perturbations.** Under larger pose and scale changes, our SIM(3)-equivariant model maintains completion quality, whereas competing methods degrade.

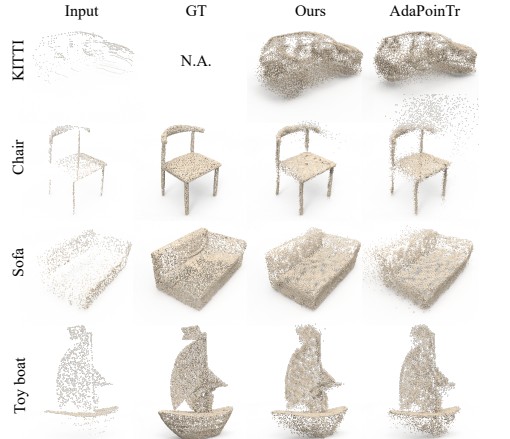

Figure 7: **Cross-domain generalization to real scans.** Our PCN-trained model completes driving (KITTI) and indoor (OmniObject3D) scans, with more details than the augmented baseline.

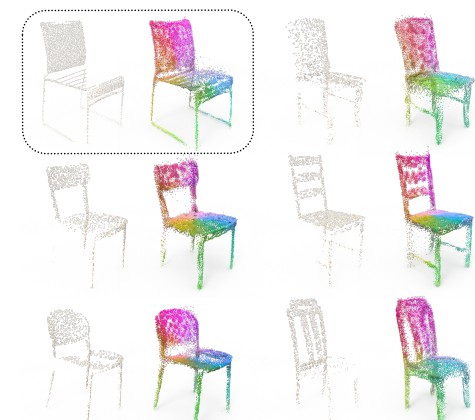

Figure 8: **Feature consistency.** Despite significant SIM(3) variations, feature maps from the PCN sample (outlined) and OmniObject3D scans exhibit matching structural patterns.

**Unseen indoor scans (OmniObject3D).** Table 3 presents cross-domain results on the diverse OmniObject3D benchmark. Our model achieves the *lowest* average CD–$\ell_1$ and *highest* F1. Compared to the top non-equivariant baseline, we reduce CD–$\ell_1$ by 14% and increase F1 by 5%. Relative to the SE(3)-equivariant ESCAPE, we achieve a 17% reduction in CD–$\ell_1$. These gains hold across all seven categories, with particularly notable improvements on *Cabinet* (14.69 *vs.* 17.15) and *Lamp* (11.07 *vs.* 14.03). Aside from augmentation-based methods, ours is the *only*

Table 2: **Cross-domain performance on KITTI.** All methods are trained on PCN Cars.

| Method | Transform | Fidelity ↓ | MMD ↓ |
|---|---|---|---|
| EquivPCN [13] | I/SO(3) | 0.413 | 3.293 |
| SCARP [15] | SE(3)/SE(3) | 2.733 | 10.020 |
| ESCAPE [10] | I/SE(3) | 1.810 | 5.930 |
| PoinTr [6] | SIM(3)/SIM(3) | **0.000** | 6.929 |
| AdaPoinTr [7] | SIM(3)/SIM(3) | 0.537 | 6.468 |
| **SIMECO** (ours) | I/SIM(3) | 0.558 | **5.353** |

model that generalizes without bounding-box normalization. Fig. 7 shows that our completions better preserve intricate geometric details.

Table 3: **Cross-domain performance on OmniObject3D.** Our model outperforms $SO(3)$- and $SE(3)$-equivariant methods and non-equivariant baselines trained with $SIM(3)$ augmentation.

| Method | Transform | Airpl. | Cab. | Car | Chair | Lamp | Sofa | Wat. | CD–$\ell_1$ ↓ | F1 ↑ |
|---|---|---|---|---|---|---|---|---|---|---|
| EquivPCN [13] | I/SO(3) | 12.05 | 16.06 | 13.91 | 13.68 | 16.84 | 13.47 | 12.67 | 14.10 | 0.543 |
| SCARP [15] | SE(3)/SE(3) | 37.40 | 56.64 | 44.79 | 45.99 | 70.24 | 38.66 | 59.50 | 50.46 | 0.106 |
| ESCAPE [10] | I/SE(3) | 9.89 | 15.79 | 11.87 | 7.78 | 19.49 | 11.12 | 10.41 | 12.34 | 0.679 |
| PoinTr [6] | SIM(3)/SIM(3) | 12.11 | 26.40 | 18.98 | 12.48 | 25.38 | 16.83 | 14.81 | 18.14 | 0.515 |
| AdaPoinTr [7] | SIM(3)/SIM(3) | 11.48 | 17.15 | 12.10 | 7.44 | 14.03 | 10.73 | 10.38 | 11.90 | 0.664 |
| **SIMECO** (ours) | I/SIM(3) | **11.20** | **14.69** | **10.10** | **6.44** | **11.07** | **9.11** | **9.12** | **10.25** | **0.698** |

**Feature visualization.** Fig. 8 compares feature maps for a PCN sample alongside those from several OmniObject3D scans under different $SIM(3)$ transforms. Despite large variations in pose and scale, the feature maps share strikingly similar structures, demonstrating that our network learns pose- and scale-invariant features that generalize effectively to real-world data.

### 4.4 Ablations and analyses

**Can pose estimation replace equivariance?** To assess whether an explicit pose estimator can substitute for built-in equivariance, we prepend ConDor [36], a state-of-the-art self-supervised $SE(3)$ pose canonicalizer (no equivalent exists for $SIM(3)$ to our knowledge), to two non-equivariant baselines (PoinTr and AdaPoinTr) on the PCN dataset. As shown in Table 4, ConDor + AdaPoinTr still fails to match the augmentation-based baseline (Table 1), and yields a CD–$\ell_1$ 15% higher and an F1 score 3% lower than our $SIM(3)$-equivariant model; ConDor + PoinTr performs even worse. In contrast, our approach achieves superior accuracy without any explicit pose estimation, demonstrating that architectural equivariance is a more effective design choice.

Table 4: **Pose estimation *vs.* equivariance.** Our model ourperforms baselines with pose estimator on PCN.

| Method | Transform | CD-$\ell_1$ ↓ | F1 ↑ |
|---|---|---|---|
| ConDor [36] + PoinTr [6] | SE(3)/SE(3) | 18.56 | 0.408 |
| ConDor [36] + AdaPoinTr [7] | SE(3)/SE(3) | 9.92 | 0.692 |
| **SIMECO** (ours) | I/SIM(3) | **8.59** | **0.714** |

Table 5: **Sensitivity to training-time transforms** on PCN Car.

| Transform | CD-$\ell_1$ ↓ | F1 ↑ |
|---|---|---|
| SIM(3)/SIM(3) | 8.88 | 0.705 |
| SE(3)/SIM(3) | 8.81 | 0.707 |
| SO(3)/SIM(3) | 8.78 | 0.708 |
| I/SIM(3) | **8.76** | **0.712** |

**How much equivariance is necessary?** Figure 9 plots performance on the PCN Car subset as we progressively swap non-equivariant for $SIM(3)$-equivariant layers in the encoder/decoder. The non-equivariant baseline [0/0] yields the worst CD–$\ell_1$ and lowest F1. Equipping the encoder with six equivariant layers [6/0] reduces CD-$\ell_1$ and boosts F1. Further introducing four equivariant decoder layers [6/4] yields a slight performance drop, likely due to the overhead from mixing equivariant and non-equivariant modules. Nevertheless, the fully equivariant setup [6/8] attains the best CD-$\ell_1$ and F1, confirming that preserving *end-to-end* $SIM(3)$ symmetry is essential for optimal shape completion.

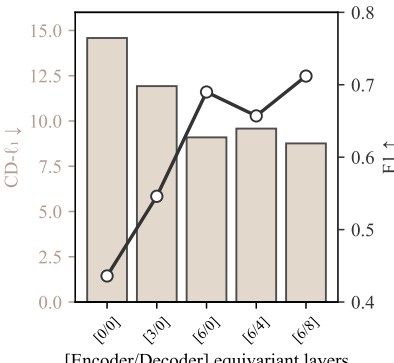

Figure 9: **Equivariant layers ablation.** On PCN Car, performance increases as non-equivariant layers are progressively replaced with $SIM(3)$-equivariant ones in the encoder/decoder. The fully equivariant setup delivers the best results.

**Which equivariance group matters most?** We ablate training-time equivariance on the PCN Car subset across four operational design domains (ODD): rotation (R), rotation + translation (R + T), rotation + scale (R + S), and full $SIM(3)$ (R + S + T). When directly transferred to KITTI scans (Fig. 10), only the $SIM(3)$ model attains the best Fidelity and MMD. Omitting scale (R + T) or translation (R + S) equivariance increases errors

on both metrics. Notably, the gap between $\mathrm{SIM}(3)$ and its subgroups highlights how much more challenging full $\mathrm{SIM}(3)$ equivariance is compared to the subgroups. This synthetic-to-real analysis confirms that $\mathrm{SIM}(3)$ equivariance is essential for robust, in-the-wild shape completion and further validates our model's advantage over methods limited to $\mathrm{SO}(3)$ or $\mathrm{SE}(3)$.

**Must we canonicalize training data?**   Real-world data seldom provide objects in a common reference frame, so a practical completion model should tolerate arbitrary pose and scale at training time as well. We therefore *train* our network on the PCN Car subset under four configurations, I, $\mathrm{SO}(3)$, $\mathrm{SE}(3)$, and $\mathrm{SIM}(3)$, and report test performance in Table 5. Across these settings, the average CD-$\ell_1$ varies by less than 0.15 and the F-score by at most 0.007. The negligible difference demonstrates that our $\mathrm{SIM}(3)$-equivariant architecture learns shape priors that are robust to the transform of training data. Thus, explicit canonicalization of the training data is *not* required.

**Robustness to input noise and point dropout.**   Our model, trained exclusively on clean PCN data, degrades gracefully under Gaussian noise up to 0.5% of the object scale, with average CD-$\ell_1$ rising modestly from 8.59 to 9.34 (Fig. 11). It also tolerates substantial point dropout: with a 25% additional dropout rate, F1 remains above 0.69 while CD-$\ell_1$ stays near its drop-free level. These results demonstrate the robustness of our architecture to real-world scans subject to noise and sparsity. We provide additional analyses in Appendix B.

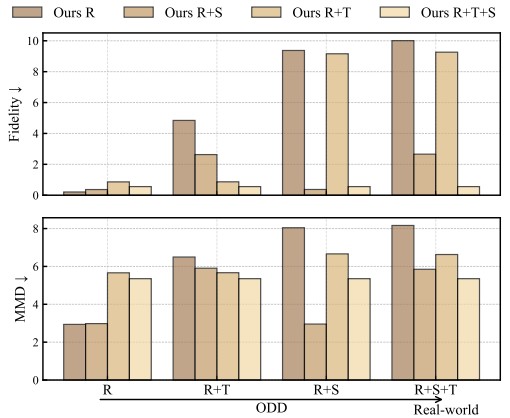

Figure 10: **Equivariance group ablation.** PCN-trained models evaluated directly on KITTI are endowed with equivariance to rotation (R), translation (T), and scale (S). Each added symmetry group improves performance, with the full $\mathrm{SIM}(3)$ model performing best in real-world ODD.

Figure 11: **Robustness to noise and dropout.** Shaded regions show category-wise min-max. Trained solely on clean PCN data, our model remains robust under increasing Gaussian noise and up to 25% additional point dropout.

# 5   Conclusion

We identified $\mathrm{SIM}(3)$ equivariance as essential for tackling the persistent pose and scale bias in shape completion and achieving robust generalization. To this end, we introduced the first shape completion architecture composed of inherently $\mathrm{SIM}(3)$-equivariant modules, which effectively disentangle intrinsic geometry from extrinsic transforms. Under a strict, unbiased evaluation protocol that removes all alignment cues, our method sets a new state of the art both on synthetic benchmarks and in direct transfer to unconstrained real scans. These results confirm architectural $\mathrm{SIM}(3)$ equivariance as a principled remedy for truly generalizable shape completion. While our current implementation is limited to single-shape completion, extending this framework to multi-object and large-scale scene modeling opens compelling avenues for future work.

# Acknowledgment

This work was supported by TUM Georg Nemetschek Institute under the AI4TWINNING project. We thank Burak Bekci for sharing pretrained ESCAPE weights, Miaowen Dong for helpful discussions, and anonymous reviewers for their constructive comments.

# A    Reproducibility

The code repository and demo are publicly accessible via the project page[1]. Detailed instructions for setup and running the code are described in the repository's `README.md` file.

# B    Additional Analysis

## B.1    Input normalization

Table 6 compares two common scale normalization schemes: per-scan bounding-box extents and the global category maximum. Both schemes are consistently applied at training and testing. We observe that the non-equivariant AdaPoinTr [7] is sensitive to the choice and performs better with bounding-box normalization, which we adopt for all competing methods in Sec. 4.2. Our model outperforms AdaPoinTr by a substantial margin, with only marginal improvements when ground-truth extents are available.

Table 6: **Effect of input scale normalization.** "B. box" scales each scan by its bounding box extent, while "max" uses the category's global maximum extent.

| Method | Source | Extent | CD$-\ell_1$ ↓ | F1 ↑ |
|---|---|---|---|---|
| AdaPoinTr [7] | Input | B. box | 9.97 | 0.629 |
| AdaPoinTr [7] | Input | Max | 12.46 | 0.557 |
| SIMECO (ours) | Input | B. box | 8.88 | 0.705 |
| SIMECO (ours) | GT | B. box | 8.76 | 0.712 |

## B.2    VN-SPD constraint

Imposing the VN-SPD constraint [34] on linear weights yields a more principled optimization than centering-based VN networks, since the center is learned rather than fixed. We validate this with an ablation that replaces VN-SPD with standard VN layers plus centering and normalization on partial inputs. As shown in Table 7, our full model clearly outperforms the centering variant.

Table 7: **Effect of VN-SPD constraint** [34]. The constraint yields a more principled optimization than the centering-based variant.

| Method | Transform | Airpl. | Cab. | Car | Chair | Lamp | Sofa | Table | Wat. | CD-$\ell_1$ ↓ | F1 ↑ |
|---|---|---|---|---|---|---|---|---|---|---|---|
| AdaPoinTr [7] | SIM(3)/SIM(3) | 6.46 | 12.17 | 10.51 | 10.29 | 7.59 | 12.26 | 8.90 | 8.14 | 9.54 | 0.661 |
| SIMECO (centered) | I/SIM(3) | 6.10 | 11.77 | 10.24 | 10.05 | 7.50 | 12.16 | 8.57 | 8.10 | 9.31 | 0.673 |
| SIMECO (ours) | I/SIM(3) | 6.02 | 10.75 | 9.27 | 9.25 | 6.66 | 11.16 | 7.82 | 7.77 | 8.59 | 0.714 |

## B.3    Computational efficiency

With a batch size of 40, SIMECO trains at about 1 hour per epoch on two NVIDIA A40 GPUs. Fig. 12 reports training losses and validation metrics of SIMECO and AdaPoinTr [7]. SIMECO converges much faster in terms of epochs. AdaPoinTr needs about 140 epochs to reduce CD-$\ell_1$ below 10, whereas our model does so in only 50 epochs.

Table 8 reports the per-scan latency on PCN. Our model processes a scan in 76 ms end-to-end, about twice as fast as the next-quickest equivariant competitor, ESCAPE [10] (148 ms), and more than twice as fast as EquivPCN [13] (172 ms) and SCARP [15] (172 ms). ESCAPE and SCARP spend extra time in post-processing alignment steps, which inflate total latency beyond the raw inference cost. AdaPoinTr remains faster at 16 ms but achieves this speed without any built-in equivariance. Overall, our method offers the best combination of speed and high-level SIM(3) symmetry preservation.

To isolate the effect of built-in SIM(3) equivariance from model capacity, we compare methods under a similar parameter budget on PCN (Table 9). When scaled to a comparable parameter count, AdaPoinTr improves over its smaller variant yet still trails our model by 0.34 in CD-$\ell_1$, suggesting that the advantage of equivariance persists after controlling for parameter count.

---

[1]https://sime-completion.github.io

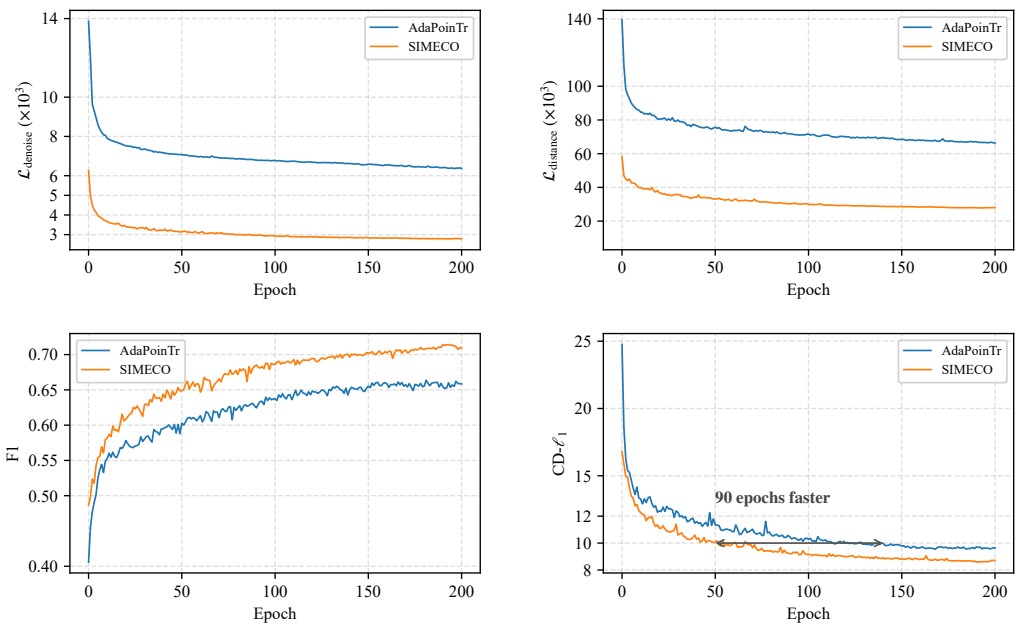

Figure 12: **Training losses and validation metrics.** Our model converges faster than the baseline.

Table 8: **Average per-scan latency on PCN.** "Inference" denotes the network forward time, measured on an NVIDIA A40 GPU; "Total" adds any post-processing overhead.

| Method | Equivariance | Latency (ms) | |
|---|---|---|---|
| | | Inference | Total |
| AdaPoinTr [7] | I | 16.4 | 16.4 |
| EquivPCN [13] | SO(3) | 171.6 | 171.6 |
| SCARP [15] | SE(3) | 159.2 | 172.0 |
| ESCAPE [10] | SE(3) | 18.1 | 148.4 |
| SIMECO (ours) | SIM(3) | 76.4 | 76.4 |

Table 9: **PCN results by parameter count.** "#Param. (M)" is the number of parameters (millions).

| Method | Transform | #Param. (M) | CD–$\ell_1$ ↓ | F1 ↑ |
|---|---|---|---|---|
| AdaPoinTr [7] | SIM(3)/SIM(3) | 32.49 | 9.54 | 0.661 |
| AdaPoinTr [7] | SIM(3)/SIM(3) | 57.12 | 8.93 | 0.693 |
| SIMECO (ours) | I/SIM(3) | 56.96 | 8.59 | 0.714 |

## B.4 Performance on thin structures

To evaluate performance on shapes with pronounced thin structures, we compute a local PCA-based anisotropy score: $L = \frac{\sigma_1 - \sigma_2}{\sigma_1}$ on each point's $k$-nearest neighbors ($k = 30$), where $\sigma_1 \geq \sigma_2 \geq \sigma_3$ are the singular values of the local covariance matrix. A high $L$ indicates a spindly, edge-like neighborhood. We then select shapes in which more than $0.5\%$ of points satisfy $L > 0.8$, yielding 50 thin-structure cases out of 1200 total. Quantitatively, on this subset, our method achieves an average CD-$\ell 1$ of 6.83, compared to 8.59 over the entire test set, demonstrating even better overall performance on such samples with thin structures. Moreover, qualitative examples such as chair and table legs in Fig. 5 and Fig. 7 further confirm that our method preserves fine details effectively.

## B.5 Limitations

Despite its strengths, our approach has several limitations:

1. **Pose- and scale-dependent features.** By construction, we remove any dependence on absolute pose or scale. While this makes the model robust to arbitrary similarity transforms, it can also discard helpful cues when objects always appear in a canonical frame. For instance, a chair back with no visible legs might be mistaken for a sofa because the two shapes coincide under a similarity transform (see Fig. 13). Nevertheless, in realistic settings our method consistently outperforms non-equivariant baselines.

2. **Symmetries across partial observations.** The equivariance property in our framework is defined with respect to a single partial scan. For different partial observations of the same object, initialization variability cannot be fully eliminated, thus cross-view symmetries cannot be explicitly enforced and must be learned implicitly from data.

3. **Articulated complex scenes.** Our method excels at completing shapes under arbitrary similarity transforms, but it does not explicitly account for independently moving sub-parts (*e.g.*, human joints, robotic arms, or scenes with multiple objects). Incorporating category-specific shape priors or allowing multiple local transforms would be natural extensions to address these more complex scenarios.

4. **Computational overhead.** Vector-valued features and fully equivariant modules incur substantial computation by a factor of three compared to scalar-valued layers. As a result, runtime latency is higher than that of non-equivariant baselines (see Table 8), which may limit real-time or resource-constrained deployments.

## B.6   Failure cases

Fig. 13 shows two failure cases that rarely occurred in our experiments. Ambiguous partial geometry can entice the network to produce a completion that is plausible yet incorrect. Severe sparsity or noise may disrupt the transform restoration and cause the completed shape to drift from the input frame.

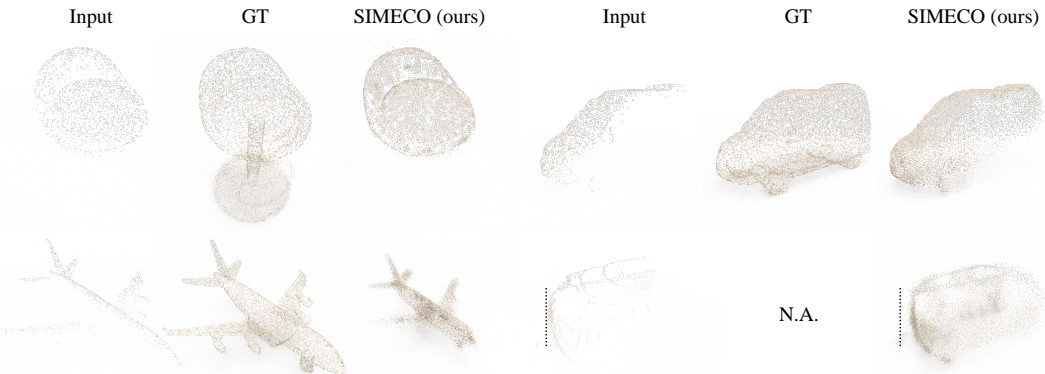

Figure 13: **Failure cases.** Top: ambiguous partial scans leave the network unsure how to complete the shape. Bottom: poor input quality disrupts the transform restoration module and yields misalignment.

## C   Proof of $\mathrm{SIM}(3)$ Equivariance

In this section, for convenience, we represent 3D vectors as row vectors and stack them into matrices. Specifically, let $\mathbf{V} \in \mathbb{R}^{D \times 3}$ denote a vector feature, where $\mathbf{V}[d] \in \mathbb{R}^3$ is its $d$-th channel, and collect $M$ such features in the set $\mathcal{V} = \{\mathbf{V}_i\}_{i=1}^{M}$. Broadcast vectors (*e.g.*, $\mathbf{1}_D = [1, 1, \ldots, 1]^\top \in \mathbb{R}^{D \times 1}$) are column vectors.

### C.1   Definitions

**Definition 1** (Group invariance)**.** *A mapping $f$ is $G$-invariant (e.g., $\mathrm{SIM}(3)$-invariant) if it satisfies $f(g \cdot \mathbf{x}) = f(\mathbf{x})$ for all $g \in G$ and admissible inputs $\mathbf{x}$.*

**Definition 2** (Group equivariance)**.** *A mapping $f$ is $G$-equivariant (e.g., $\mathrm{SIM}(3)$-equivariant) if it satisfies $f(g \cdot \mathbf{x}) = g \cdot f(\mathbf{x})$ for all $g \in G$ and admissible inputs $\mathbf{x}$.*

## C.2 SIM(3)-equivariant vector neurons

We assume the input is transformed by an arbitrary $g = (s, R, t) \in \text{SIM}(3)$, acting on each 3D vector in the VN architecture [32] and the associated matrix as follows:

$$g \cdot \mathbf{V}[d] = sR\mathbf{V}[d] + t, \qquad g \cdot \mathbf{V} = s\mathbf{V}R + \mathbf{1}_D\, t, \qquad s \in \mathbb{R}+,\ R \in \text{SO}(3),\ t \in \mathbb{R}^3. \qquad (10)$$

**VN-Linear.** The VN-Linear layer is defined as a linear transformation shared across the three columns of $\mathbf{V}$:

$$\text{VN-Linear}(\mathbf{V}) = \mathbf{W}\mathbf{V}, \quad \mathbf{W} \in \mathbb{R}^{D' \times D}. \qquad (11)$$

For translation equivariance, we constrain each row of the weight matrix $\mathbf{W}$ to sum to one [34]:

$$\sum_{j=1}^{D} w_{ij} = 1,\ \ w_{ij} \in \mathbf{W} \quad \forall i \in \{1, \ldots, D'\}, \quad \Longleftrightarrow \quad \mathbf{W}\,\mathbf{1}_D = \mathbf{1}_{D'}. \qquad (12)$$

**Proposition 1.** VN-Linear$(\cdot)$ *is* $\text{SIM}(3)$*-equivariant.*

*Proof.* For all $g \in \text{SIM}(3)$,

$$\text{VN-Linear}(g \cdot \mathbf{V}) = \mathbf{W}(s\mathbf{V}R + \mathbf{1}_D\, t) \qquad (13)$$
$$= s\mathbf{W}\mathbf{V}R + \mathbf{W}\,\mathbf{1}_D\, t \qquad (14)$$
$$= s(\mathbf{W}\mathbf{V})R + \mathbf{1}_{D'}\, t \qquad (15)$$
$$= s\,\text{VN-Linear}(\mathbf{V})R + \mathbf{1}_{D'}\, t \qquad (16)$$
$$= g \cdot \text{VN-Linear}(\mathbf{V}). \qquad (17)$$
$$\square$$

**VN-ReLU.** The VN-ReLU layer is constructed via three VN-Linear layers that produce a feature $\mathbf{F}$, a direction $\mathbf{B}$, and an origin $\mathbf{O}$, followed by centering with respect to $\mathbf{O}$:

$$(\mathbf{F}, \mathbf{B}, \mathbf{O}) := \text{VN-Linear}(\mathbf{V}), \quad \mathbf{F_O} = \mathbf{F} - \mathbf{O}, \quad \mathbf{B_O} = \mathbf{B} - \mathbf{O}. \qquad (18)$$

The nonlinearity removes the negative projection of $\mathbf{F_O}$ onto the normal $\mathbf{B_O}$ of the plane through $\mathbf{O}$:

$$\text{VN-ReLU}(\mathbf{V}) = \begin{cases} \mathbf{O} + \mathbf{F_O} & \text{if } \langle \mathbf{F_O}, \mathbf{B_O} \rangle_F \geq 0 \\ \mathbf{O} + \mathbf{F_O} - \left\langle \mathbf{F_O}, \dfrac{\mathbf{B_O}}{\|\mathbf{B_O}\|_2} \right\rangle_F \dfrac{\mathbf{B_O}}{\|\mathbf{B_O}\|_2} & \text{o.w.} \end{cases}$$

$$= \begin{cases} \mathbf{F} & \text{if } \langle \mathbf{F_O}, \mathbf{B_O} \rangle_F \geq 0 \\ \mathbf{F} - \langle \mathbf{F_O}, \mathbf{B_O} \rangle_F \dfrac{\mathbf{B_O}}{\|\mathbf{B_O}\|_2^2} & \text{o.w.} \end{cases} . \qquad (19)$$

**Proposition 2.** VN-ReLU$(\cdot)$ *is* $\text{SIM}(3)$*-equivariant.*

*Proof.* For all $g \in \text{SIM}(3)$,

$$\text{VN-ReLU}(g \cdot \mathbf{V}) \overset{(*)}{=} \begin{cases} g \cdot \mathbf{F} & \text{if } \langle s\mathbf{F_O}R, s\mathbf{B_O}R \rangle_F \geq 0 \\ s\mathbf{F}R + \mathbf{1}_D\, t - \langle s\mathbf{F_O}R, s\mathbf{B_O}R \rangle_F \dfrac{s\mathbf{B_O}R}{\|s\mathbf{B_O}R\|_2^2} & \text{o.w.} \end{cases} \qquad (20)$$

$$\overset{(**)}{=} \begin{cases} g \cdot \mathbf{F} & \text{if } s^2 \langle \mathbf{F_O}, \mathbf{B_O} \rangle_F \geq 0 \\ s\mathbf{F}R + \mathbf{1}_D\, t - s^2 \langle \mathbf{F_O}, \mathbf{B_O} \rangle_F \dfrac{s\mathbf{B_O}R}{s^2\|\mathbf{B_O}\|_2^2} & \text{o.w.} \end{cases} \qquad (21)$$

$$= \begin{cases} g \cdot \mathbf{F} & \text{if } \langle \mathbf{F_O}, \mathbf{B_O} \rangle_F \geq 0 \\ s\mathbf{F}R + \mathbf{1}_D\, t - s \langle \mathbf{F_O}, \mathbf{B_O} \rangle_F \dfrac{\mathbf{B_O}}{\|\mathbf{B_O}\|_2^2}R & \text{o.w.} \end{cases} \qquad (22)$$

$$= \begin{cases} g \cdot \mathbf{F} & \text{if } \langle \mathbf{F_O}, \mathbf{B_O} \rangle_F \geq 0 \\ s\big(\mathbf{F} - \langle \mathbf{F_O}, \mathbf{B_O} \rangle_F \dfrac{\mathbf{B_O}}{\|\mathbf{B_O}\|_2^2}\big)R + \mathbf{1}_D\, t & \text{o.w.} \end{cases} \qquad (23)$$

$$= \begin{cases} g \cdot \mathbf{F} & \text{if } \langle \mathbf{F_O}, \mathbf{B_O} \rangle_F \geq 0 \\ g \cdot \big(\mathbf{F} - \langle \mathbf{F_O}, \mathbf{B_O} \rangle_F \dfrac{\mathbf{B_O}}{\|\mathbf{B_O}\|_2^2}\big) & \text{o.w.} \end{cases} \qquad (24)$$

$$= g \cdot \text{VN-ReLU}(\mathbf{V}). \qquad (25)$$

Here, $(*)$ holds because $\mathbf{F}, \mathbf{B}, \mathbf{O}$ are SIM(3)-equivariant (Prop. 1), and translation cancels in $\mathbf{F_O}$ and $\mathbf{B_O}$. $(**)$ holds as the Frobenius inner product and the $\ell_2$-norm are rotation-invariant. $\qquad \square$

**VN-LeakyReLU.** The VN-LeakyReLU layer is a minor variant of VN-ReLU:

$$\text{VN-LeakyReLU}(\mathbf{V}) = \alpha \mathbf{V} + (1-\alpha)\,\text{VN-ReLU}(\mathbf{V}), \quad \alpha \in (0,1). \tag{26}$$

This operation is trivially SIM(3)-equivariant.

**VN-Max.** The VN-Max layer is defined on a set of vector features $\mathcal{V}$ by applying two shared VN-Linear layers to each $\mathbf{V}_i \in \mathcal{V}$, producing a direction $\mathbf{B}_i$ and an origin $\mathbf{O}_i$, followed by centering with respect to $\mathbf{O}_i$:

$$(\mathbf{B}_i, \mathbf{O}_i) \coloneqq \text{VN-Linear}(\mathbf{V}_i), \quad \mathbf{B}_{\mathbf{O},i} = \mathbf{B}_i - \mathbf{O}_i, \quad \mathbf{V}_{\mathbf{O},i} = \mathbf{V}_i - \mathbf{O}_i. \tag{27}$$

VN-Max selects, for each channel $d$, the feature whose centered representation $\mathbf{V}_{\mathbf{O},i}[d]$ is most aligned with its corresponding centered direction $\mathbf{B}_{\mathbf{O},i}[d]$:

$$\text{VN-Max}(\mathcal{V})[d] = \mathbf{V}_{i^*}[d], \quad \text{with}\ \ i^* = \arg\max_i \langle \mathbf{V}_{\mathbf{O},i}[d], \mathbf{B}_{\mathbf{O},i}[d] \rangle_F. \tag{28}$$

**Proposition 3.** VN-Max$(\cdot)$ *is* SIM(3)*-equivariant.*

*Proof.* For all $g \in \text{SIM}(3)$,

$$\text{VN-Max}(g \cdot \mathcal{V})[d] = g \cdot \mathbf{V}_{i^*}[d], \quad \text{with}\ \ i^* = \arg\max_i \langle s\mathbf{V}_{\mathbf{O},i}[d]R, s\mathbf{B}_{\mathbf{O},i}[d]R \rangle_F \tag{29}$$

$$\overset{(*)}{=} \arg\max_i s^2 \langle \mathbf{V}_{\mathbf{O},i}[d], \mathbf{B}_{\mathbf{O},i}[d] \rangle_F \tag{30}$$

$$\overset{(**)}{=} \arg\max_i \langle \mathbf{V}_{\mathbf{O},i}[d], \mathbf{B}_{\mathbf{O},i}[d] \rangle_F \tag{31}$$

$$\text{VN-Max}(g \cdot \mathcal{V})[d] = g \cdot \text{VN-Max}(\mathcal{V})[d] \quad \Longleftrightarrow \quad \text{VN-Max}(g \cdot \mathcal{V}) = g \cdot \text{VN-Max}(\mathcal{V}). \tag{32}$$

Here, $(*)$ holds since the Frobenius inner product is rotation-invariant. $(**)$ holds as the positive scaling factor $s^2$ preserves the ordering, so the index $i^*$ remains unchanged. $\qquad \square$

### C.3 SIM(3)-equivariant Transformer

**Canonicalization.** VN-LayerNorm follows the definition in Sec. 3.2:

$$\mathbf{V}' = \text{VN-LayerNorm}(\mathbf{V}) = \text{layernorm}\left(\left\|\mathbf{V} - \bar{\mathbf{V}}\right\|_2\right) \cdot \frac{\mathbf{V} - \bar{\mathbf{V}}}{\left\|\mathbf{V} - \bar{\mathbf{V}}\right\|_2}, \quad \text{with}\ \ \bar{\mathbf{V}} = \frac{1}{D}\sum_{d=1}^{D} \mathbf{V}[d]. \tag{33}$$

**Proposition 4.** VN-LayerNorm$(\cdot)$ *is invariant to scaling and translation, and equivariant to rotation.*

*Proof.* For all $g \in \text{SIM}(3)$,

$$\text{VN-LayerNorm}(g \cdot \mathbf{V}) \overset{(*)}{=} \text{layernorm}\left(\left\|s\mathbf{V}R - s\bar{\mathbf{V}}R\right\|_2\right) \cdot \frac{(s\mathbf{V}R - s\bar{\mathbf{V}}R)}{\left\|s\mathbf{V}R - s\bar{\mathbf{V}}R\right\|_2} \tag{34}$$

$$\overset{(**)}{=} \text{layernorm}\left(s\left\|\mathbf{V} - \bar{\mathbf{V}}\right\|_2\right) \cdot \frac{\mathbf{V} - \bar{\mathbf{V}}}{\left\|\mathbf{V} - \bar{\mathbf{V}}\right\|_2}R \tag{35}$$

$$\overset{(***)}{=} \text{layernorm}\left(\left\|\mathbf{V} - \bar{\mathbf{V}}\right\|_2\right) \cdot \frac{\mathbf{V} - \bar{\mathbf{V}}}{\left\|\mathbf{V} - \bar{\mathbf{V}}\right\|_2}R \tag{36}$$

$$= \text{VN-LayerNorm}(\mathbf{V})R. \tag{37}$$

$(*)$ holds since $\bar{\mathbf{V}}$ is SIM(3)-equivariant, and translation cancels in differences. $(**)$ holds because the $\ell_2$-norm is rotation-invariant. $(***)$ holds as layer normalization is invariant to positive scaling. $\qquad \square$

**Shape reasoning.** VN-Attn follows the definition in Sec. 3.2. For self-attention, the input features satisfy $\mathbf{V}'_q = \mathbf{V}'_k = \mathbf{V}'$; for cross-attention, $\mathbf{V}'_q$ and $\mathbf{V}'_k$ may differ. These features are the outputs of the canonicalization step, which removes the effects of translation and scale. The query and key are computed via shared VN-Linear layers:

$$\mathbf{Q}_i := \text{VN-Linear}(\mathbf{V}'_{q,i}), \quad \mathbf{K}_j := \text{VN-Linear}(\mathbf{V}'_{k,j}). \tag{38}$$

The attention weight and output are then computed following VN-Transformer [33]:

$$a_{i,j} = \text{VN-Attn}(\mathbf{Q}_i, \mathbf{K}_j) = \text{softmax}_j \left( \frac{1}{\sqrt{3D}} \langle \mathbf{Q}_i, \mathbf{K}_j \rangle_F \right), \tag{39}$$

$$\mathbf{Z}_i = \sum_j a_{i,j} \cdot \text{VN-Linear}(\mathbf{V}'_{k,j}). \tag{40}$$

**Proposition 5.** VN-Attn$(\cdot, \cdot)$ *is invariant to rotation, and* $\mathbf{Z}_i$ *is equivariant to rotation.*

*Proof.* Rotation invariance of VN-Attn$(\cdot, \cdot)$ follows immediately from the fact that the Frobenius inner product is rotation-invariant. Because $a_{i,j}$ is rotation-invariant and, by Prop. 1, VN-Linear$(\cdot)$ is rotation-equivariant, it follows that $\mathbf{Z}_i$ is rotation-equivariant. □

**Transform restoration.** Transform restoration follows the definition in Sec. 3.2. Given the module input $\mathbf{V}$ and the attention output $\mathbf{Z}$, the restored output is then computed as

$$\text{TR}(\mu, \mathbf{V}, \mathbf{Z}) = \mathbf{V} + \text{VN-Linear}(\mu\mathbf{Z}), \quad \text{with } \mu = \mathbb{E}_D \big\| \mathbb{E}_i(\mathbf{V}_i - \bar{\mathbf{V}}_i) \big\|_2, \; \bar{\mathbf{V}}_i = \frac{1}{D} \sum_{d=1}^{D} \mathbf{V}_i[d]. \tag{41}$$

**Proposition 6.** TR$(\cdot, \cdot, \cdot)$ *can recover* SIM(3) *equivariance.*

*Proof.* For all $g \in \text{SIM}(3)$

$$\text{TR}(g \cdot (\mu, \mathbf{V}, \mathbf{Z})) \overset{(*)}{=} s\mathbf{V}R + \mathbf{1}_D\, t + \text{VN-Linear}(s\mu\mathbf{Z}R) \tag{42}$$

$$\overset{(**)}{=} s\mathbf{V}R + \mathbf{1}_D\, t + s\mu\text{VN-Linear}(\mathbf{Z})R \tag{43}$$

$$= s(\mathbf{V} + \mu\text{VN-Linear}(\mathbf{Z}))R + \mathbf{1}_D\, t \tag{44}$$

$$= g \cdot \text{TR}((\mu, \mathbf{V}, \mathbf{Z})) \tag{45}$$

Here, $(*)$ holds because the attention output $\mathbf{Z}$ encodes only the effect of rotation (Prop. 4 and Prop. 5). The scalar $\mu$ scales with $s$, as translation is eliminated by differencing, and the $\ell_2$-norm is rotation-invariant. Hence,

$$\mathbb{E}_D \big\| \mathbb{E}_i(g \cdot (\mathbf{V}_i - \bar{\mathbf{V}}_i)) \big\|_2 = s \cdot \mathbb{E}_D \big\| \mathbb{E}_i(\mathbf{V}_i - \bar{\mathbf{V}}_i) \big\|_2. \tag{46}$$

$(**)$ holds because VN-Linear$(\cdot)$ is SIM(3)-equivariant (Prop. 1). □

## C.4 Other modules

**VN-DGCNN.** VN-DGCNN performs edge feature extraction and aggregation across layers [7, 32]:

$$\mathbf{V}_i^{l+1} = \text{VN-Max}_{j \in \mathcal{N}_i} \left( \text{VNLA} \left( (\mathbf{V}_j^l + \bar{\mathbf{V}}^l - \mathbf{V}_i^l) \oplus \mathbf{V}_i^l \right) \right), \quad \text{with } \bar{\mathbf{V}}^l = \frac{1}{M} \sum_{i=1}^{M} \mathbf{V}_i^l. \tag{47}$$

where $\mathcal{N}_i$ is the KNN neighborhood of point $i$, and $\oplus$ denotes feature concatenation. VNLA$(\cdot)$ applies VN-Linear$(\cdot)$ followed by VN-LeakyReLU$(\cdot)$. Because each edge feature $(\mathbf{V}_j^l + \bar{\mathbf{V}}^l - \mathbf{V}_i^l) \oplus \mathbf{V}_i^l$ preserves SIM(3) equivariance, and both VNLA$(\cdot)$ and VN-Max$(\cdot)$ are SIM(3)-equivariant (Prop. 1, Prop. 2, and Prop. 3), each layer output remains equivariant. By layer-wise induction, the entire VN-DGCNN is SIM(3)-equivariant. We initialize all vector features $\mathbf{V}$ with the 3D coordinates of the input points.

**Query generator.** The query generator produces a fused query set $\mathbf{Q} = [\mathbf{Q}_I, \mathbf{Q}_G]$ [7], where $\mathbf{Q}_I$ is sampled from the partial input and $\mathbf{Q}_G = \text{VN-Linear}(\text{VN-Max}(\mathcal{V}))$, with $\mathcal{V}$ denoting the output of the final encoder layer. $\mathbf{Q}$ is $\text{SIM}(3)$-equivariant, as $\mathbf{Q}_I$ follows the transformed input, and $\mathbf{Q}_G$ inherits equivariance from the encoder through $\text{SIM}(3)$-equivariant operations (Prop. 1 and Prop. 3).

**Reconstruction head.** The reconstruction head produces the final output point set $\hat{\mathbf{y}}$ as:

$$\hat{\mathbf{y}} = \text{VN-Linear}(\mathbf{V} - \bar{\mathbf{V}}) + \mathbf{Q}, \quad \text{with} \quad \bar{\mathbf{V}} = \frac{1}{D} \sum_{d=1}^{D} \mathbf{V}[d]. \tag{48}$$

where $\mathbf{V}$ is the decoder output. $\hat{\mathbf{y}}$ is $\text{SIM}(3)$-equivariant, as centering $\mathbf{V}$ prevents translation accumulation from $\mathbf{V}$ and $\mathbf{Q}$, with both the $\text{VN-Linear}(\cdot)$ and $\mathbf{Q}$ preserving equivariance (Prop. 1).

### C.5 Summary and approximate equivariance bound

The entire network architecture is $\text{SIM}(3)$-equivariant by construction, since it is built exclusively from the above-mentioned $\text{SIM}(3)$-equivariant modules. To stabilize training, we follow the practice of Assaad *et al.* [33] and introduce a small norm-controlled bias to VN-Linear layers. Although this modification introduces a minor deviation from exact equivariance, its effect in each layer is bounded by a constant $\epsilon_l$, and remains insignificant across layers as proved in VN-Transformer [33]. As a result, the overall network is effectively $\epsilon_{1...L}$-approximately equivariant.

## D Implementation Details

SIMECO is implemented in PyTorch and optimized using the Adam optimizer with an initial learning rate of $10^{-4}$, a weight decay of $5 \times 10^{-4}$, and a learning-rate decay factor of 0.9 every 15 epochs. We adopt the same architectural depth and hyperparameters as AdaPoinTr [7]. The models, including baselines, were trained for 200 epochs on two NVIDIA A40 GPUs. All other completion methods [6, 7, 8, 10, 13, 15, 54, 64] were used with their default settings.

## E More Visualizations

Fig. 14 expands the PCN comparison with more methods. Fig. 15 presents further qualitative results on KITTI and OmniObject3D scans. Fig. 16 shows how the methods respond to controlled pose and scale perturbations.

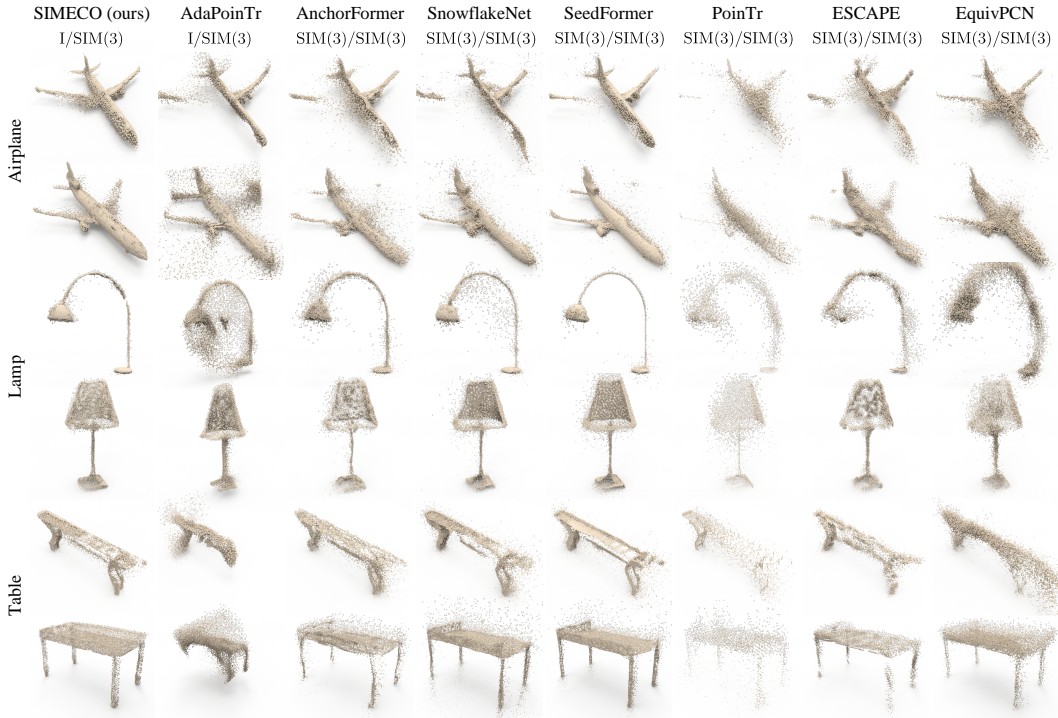

Figure 14: **Extended comparison on PCN.** Our model outperforms other equivariant methods and non-equivariant baselines trained with $\mathrm{SIM}(3)$ augmentation. Complements Fig. 5.

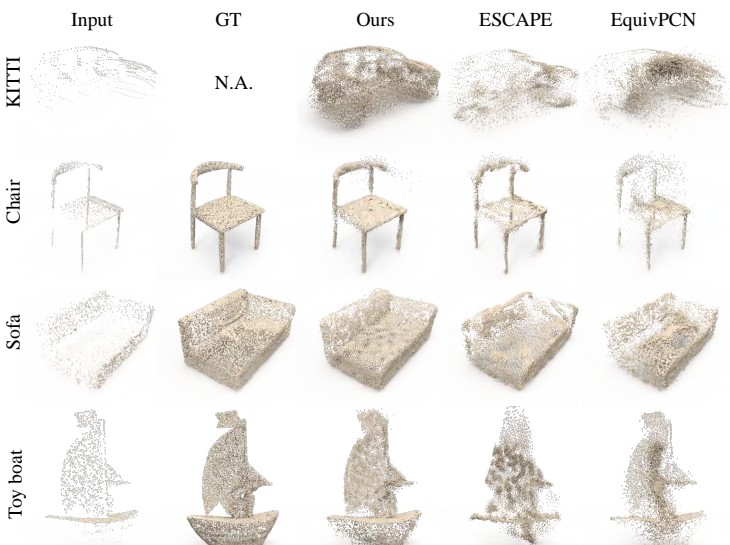

Figure 15: **Extended cross-domain comparison.** Our PCN-trained model completes driving (KITTI) and indoor (OmniObject3D) scans more accurately than other methods with $\mathrm{SIM}(3)$ augmentation. Complements Fig. 7.

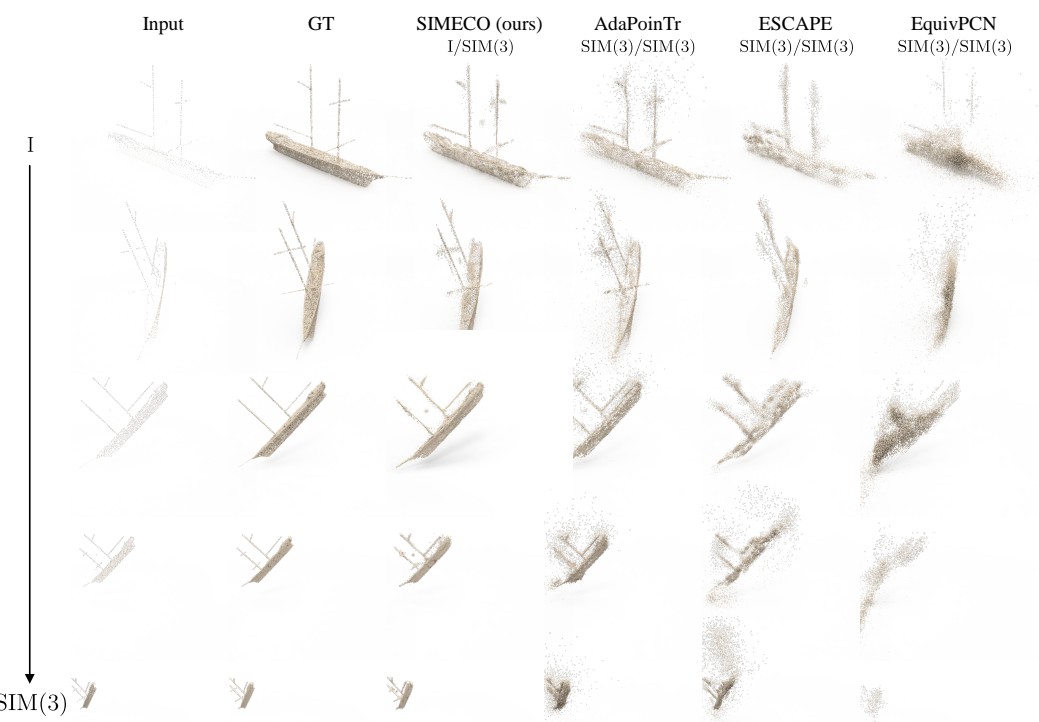

Figure 16: **Extended comparison of robustness to pose and scale perturbations.** Under larger pose and scale changes, our $SIM(3)$-equivariant model maintains completion quality, whereas competing methods degrade. Complements Fig. 6.

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
