# OpenReview forum: "Learning Generalizable Shape Completion with SIM(3) Equivariance"
_NeurIPS.cc/2025/Conference — NeurIPS 2025 poster_

### Official Review · Reviewer_1dGn · 2025-06-26

**Clarity:** 3
**Significance:** 3
**Originality:** 3
**Rating:** 5
**Confidence:** 3

**Summary:**

The paper proposes a SIM(3)-equivariant shape completion network architecture based on an adaptation of the AdaPoinTr backbone. The original Transformer layers are replaced with novel SIM(3)-equivariant modules.

The proposed SIM(3)-equivariant module consists of three subcomponents:
1. a canonicalization  module (layer norm extension) that normalizes scale and translation, while being rotation equivariant,
2. a shape reasoning module (VN-transformer) that is rotation invariant, to obtain a fully SIM(3)-invariant representation,
3. and a transform restoration module to restore scale and translation, to obtain a SIM(3)-equivariant output.

The resulting model, SIMECO, achieves significant improvements over state-of-the-art methods trained with SIM(3) data augmentation. On the PCN benchmark, it outperforms AdaPoinTr by 10% in CD-L1 and 8% in F1-score. However, SIMECO’s forward pass latency is higher (76 ms vs. 16 ms for AdaPoinTr).

**Questions:**

1. **Training Convergence** - All methods are reported to be trained for 200 epochs. However, did all of them fully converge? It would be helpful to include training and validation loss curves—especially for AdaPoinTr vs. SIMECO. I suspect that SIMECO may converge faster since it does not rely on SIM(3) data augmentation, but it's not entirely clear from the current presentation.

2. **Table 2 – Possible Typo** - In Table 2, PoinTR is shown with a Fidelity score of 0.000. Is this a typo, or is there an explanation?

3. **Clarification on Figure 10** - If I understood Figure 10 correctly, the model is trained with $N_{in}=2018$ input points, but evaluated at test time with fewer points. Is that interpretation correct?

**Ethical Concerns:**

["NO or VERY MINOR ethics concerns only"]

**Final Justification:**

The author addressed my concerns, as a consequence, I recommend accepting the paper.

**Limitations:**

The limitations are discussed in the supp. mat.

**Paper Formatting Concerns:**

/

**Quality:**

4

**Strengths And Weaknesses:**

## Strengths
1. The paper proposes the first SIM(3)-equivariant network architecture for shape completion.
2. The proposed SIM(3)-equivariant module is intuitive and simple.
3. The paper is well illustrated, especially with the use of color-coded circles and triangles to clearly highlight different types of equivariances and invariances.
4. An extensive ablation study supports the architectural and design choices.


## Weaknesses
1. The motivation for introducing the relaxed problem in Equation (4) is unclear. Since Equation (2) appears sufficient, the use of a relaxed version where $g^* \neq I$ seems to unnecessarily complicate the formulation. Clarification or justification would strengthen this point.

2. References to the supplementary material are missing in several places. Explicitly pointing to the supplement would help readers, especially since important derivations and explanations are located there.

3. Inconsistencies in notation: below Equation (9), the terms $E_{D^l}$​ and $E_i$​ appear to denote expectations, yet the standard expectation notation $\mathbb{E}$ is not used (even though it is sometimes used in the supp. mat., albeit inconsistently).

4. SIMECO is much slower (76ms vs 16ms) than AdaPoinTr and only increases the performance by 10%. The trade-off between accuracy and efficiency could be more thoroughly discussed.

---

> ### Author Rebuttal · Authors · 2025-07-28
>
> Dear Reviewer 1dGn,
>
> Thank you for your comments. We address your concerns and questions below.
>
> > The motivation for introducing the relaxed problem in Equation (4) is unclear. Since Equation (2) appears sufficient, the use of a relaxed version where seems to unnecessarily complicate the formulation. Clarification or justification would strengthen this point.
>
> We appreciate your suggestion to improve the exposition. While Eq. (2) might seem sufficient from a theoretical standpoint, it entangles geometric inference and transform alignment. To remove the bias embedded in the transform, we need to disentangle the geometry from the transform and let the network learn the former. Once we do so, the immediately inferred geometry (in a canonical frame) is invariant to the transform. Thus, to re-align the geometry to the input frame, the transform must then be restored. Accordingly, $\min_{\theta} \mathcal L\bigl(f_{\theta}(\mathbf{x}), \mathbf{y}\bigr)$ in Eq. (2) needs to be decomposed into two components: Req. (1) and Req. (3) in Eq. (4), for geometric reasoning and transform alignment, respectively. We have made the motivation behind Eq. (4) more explicit in the revised manuscript.
>
> > References to the supplementary material are missing in several places. Explicitly pointing to the supplement would help readers, especially since important derivations and explanations are located there.
>
> Thank you for the suggestion. We have added explicit references to the Suppl. Mat. at all relevant points, directing readers to the detailed derivations and explanations.
>
> > Inconsistencies in notation: below Eq. (9), the terms and appear to denote expectations, yet the standard expectation notation is not used (even though it is sometimes used in the supp. mat., albeit inconsistently).
>
> Thank you for pointing out this issue. We have revised Eq. (9) and Eq. (46) (and all related occurrences) to use the standard expectation notation $\mathbb{E}[\cdot]$ consistently throughout both the main text and the Suppl. Mat.
>
> > SIMECO is much slower (76ms vs 16ms) than AdaPoinTr and only increases the performance by 10\%. The trade-off between accuracy and efficiency could be more thoroughly discussed.
>
> While SIMECO incurs higher latency, it delivers substantial quality gains, reducing MMD on KITTI by 17\% and CD-$\ell1$ on OmniObject3D by 14\% without any data augmentation. For a fair comparison among equivariant completion methods, SIMECO is not only the fastest but also uniquely outperforms an augmented AdaPoinTr within its target equivariance group. Notably, Fig. 13 in the Suppl. Mat. demonstrates that SIMECO maintains performance under large pose and scale perturbations, whereas AdaPoinTr degrades. In real-world settings where object scales vary significantly and cannot always be normalized (e.g., due to outliers), SIMECO’s equivariance ensures more reliable and robust completions.
>
> > Training Convergence - All methods are reported to be trained for 200 epochs. However, did all of them fully converge? It would be helpful to include training and validation loss curves—especially for AdaPoinTr vs. SIMECO. I suspect that SIMECO may converge faster since it does not rely on SIM(3) data augmentation, but it's not entirely clear from the current presentation.
>
> Thank you for the suggestion. All models were trained for 200 epochs and fully converged. While the rebuttal format precludes the inclusion of rich media, we have tabulated training losses and validation metrics at 10-epoch intervals (see table below) and added the full curves to the revised manuscript. These show that SIMECO's (S) losses and metrics improve far more effectively than AdaPoinTr’s (A). For instance, AdaPoinTr requires roughly 140 epochs to drive CD-$\ell_1$ below 10, whereas SIMECO reaches the same threshold in only about 50 epochs.
>
> |Epoch|$\mathcal{L}_{\mathrm{denoise}}$↓ (A)|$\mathcal{L}_{\mathrm{denoise}}$↓ (S)|$\mathcal{L}_{\mathrm{distance}}$↓ (A)|$\mathcal{L}_{\mathrm{distance}}$↓ (S)|CD-$\ell_1$↓ (A)|CD-$\ell_1$↓ (S)|F1↑ (A)|F1↑ (S)|
> |---|---|---|---|---|---|---|---|---|
> |0|13.86|6.26|139.44|58.17|24.76|16.78|0.406|0.487|
> |10|7.90|3.67|84.64|39.84|13.27|12.26|0.552|0.581|
> |20|7.52|3.40|80.47|36.78|12.39|11.11|0.578|0.608|
> |30|7.36|3.30|79.91|35.60|12.29|10.60|0.581|0.629|
> |40|7.17|3.21|76.58|34.13|11.59|10.29|0.595|0.644|
> |50|7.07|3.16|75.67|33.10|11.35|10.09|0.603|0.648|
> |60|6.98|3.12|73.90|32.69|10.84|10.03|0.613|0.647|
> |70|6.91|3.06|73.42|31.99|11.03|9.64|0.611|0.666|
> |80|6.85|3.00|72.57|30.97|10.61|9.43|0.624|0.672|
> |90|6.80|2.97|71.86|30.34|10.47|9.20|0.625|0.685|
> |100|6.77|2.92|71.62|29.89|10.29|9.17|0.637|0.686|
> |110|6.74|2.89|71.09|29.47|10.11|8.98|0.643|0.693|
> |120|6.68|2.88|69.89|29.29|9.97|8.98|0.648|0.695|
> |130|6.65|2.87|69.44|29.06|9.93|8.95|0.649|0.697|
> |140|6.62|2.86|69.66|28.86|9.99|8.91|0.650|0.696|
> |150|6.58|2.84|68.39|28.66|9.78|8.83|0.655|0.702|
> |160|6.54|2.82|67.83|28.39|9.65|8.80|0.658|0.702|
> |170|6.48|2.81|67.30|28.28|9.70|8.74|0.657|0.707|
> |180|6.44|2.80|67.13|27.92|9.72|8.70|0.655|0.707|
> |190|6.42|2.79|67.12|27.88|9.71|8.59|0.654|0.714|
> |200|6.37|2.79|66.23|28.00|9.64|8.71|0.658|0.709|
>
> > Table 2 – Possible Typo - In Table 2, PoinTR is shown with a Fidelity score of 0.000. Is this a typo, or is there an explanation?
>
> This is not a typo—PoinTr’s fidelity score is indeed 0.000 by design. Fidelity is defined as the average nearest-neighbor distance from each input point to the output shape, so a score of zero means the model leaves every input point exactly in place. PoinTr trades off this perfect input preservation against higher generative capacity, which is why it reports zero fidelity in both the original PoinTr [1] and AdaPoinTr [2] papers. We believe that Minimal Matching Distance (MMD) is often a more intuitive metric for overall completion quality. On MMD (i.e. Chamfer distance to the closest PCN car), SIMECO outperforms PoinTr by 23\% (5.353 vs. 6.929 in Table 2), demonstrating stronger completion fidelity to typical car geometries.
>
> > Clarification on Figure 10 - If I understood Figure 10 correctly, the model is trained with N=2018 input points, but evaluated at test time with fewer points. Is that interpretation correct?
>
> Yes, your interpretation is correct, except that we train with $N_\mathrm{in}=2048$ (not 2018) input points. Our encoder (following the original DGCNN design [3], with VN modifications to ensure SIM(3) equivariance [4, 5]) aggregates point features onto a fixed set of "point proxies" via $k$-nearest neighbors. As a result, at test time the network can gracefully handle any reasonable number of input points without retraining, making it well suited for real-world scans.
>
> Thank you again for your insightful feedback. We hope our response has resolved your concerns and are happy to clarify further if needed.
>
> References:
>
> [1] Yu et al. Diverse Point Cloud Completion with Geometry-aware Transformers. ICCV 2021.
>
> [2] Yu et al. Diverse Point Cloud Completion with Adaptive Geometry-aware Transformers. TPAMI 2023.
>
> [3] Wang et al. Dynamic Graph CNN for Learning on Point Clouds. TOG 2019.
>
> [4] Deng et al. Vector Neurons: A General Framework for SO(3)-Equivariant Networks. ICCV 2021.
>
> [5] Katzir et al. Shape-Pose Disentanglement using SE(3)-Equivariant Vector Neurons. ECCV 2022.

---

> > ### Comment · Reviewer_1dGn · 2025-08-01
> >
> > I thank the authors for the rebuttal. It fully resolves my concerns (Training Convergence, Table 2 – Possible Typo and Clarification on Figure 10).

---

> > > ### Author Response · Authors · 2025-08-04
> > > **Thanks for your reply**
> > >
> > > Thank you for your prompt response. We’re glad our rebuttal fully resolved your concerns and remain available until the end of the discussion period for any further questions.

---

### Official Review · Reviewer_ztk1 · 2025-06-29

**Clarity:** 3
**Significance:** 3
**Originality:** 3
**Rating:** 4
**Confidence:** 3

**Summary:**

In this paper, to address the bias learning caused by leaked pose and scale cues in most 3D shape completion methods, the authors proposed a more robust shape completion network that ensures SIM(3)-equivariant. With this novel shape completion network, a completed 3D shape can be inferred without the need for any ground-truth alignment, which makes the proposed method more available in practice use. The main contribution of the paper is stating the bias learning caused by leaked pose and scale cues in most traditional 3D shape completion methods; meanwhile, to address this issue, the authors decompose the optimizing objective into three parts: geometric reasoning, equivariance, and transform alignment. With this designed fully SIM(3)-equivariant network for shape completion, the inference 3D shapes follow arbitrary similarity transforms applied to inputs. The experiments show the effectiveness of the proposed method.

**Questions:**

Please refer to the weakness part.

**Ethical Concerns:**

["NO or VERY MINOR ethics concerns only"]

**Final Justification:**

I thank the authors for their comprehensive responses, which have satisfactorily addressed all of my initial comments. I suggest including these discussions in the final version for completeness. I will therefore maintain my original positive rating.

**Limitations:**

Yes.

**Paper Formatting Concerns:**

No.

**Quality:**

3

**Strengths And Weaknesses:**

Strengths:
1. The motivation and idea of this paper are reasonable, and the problem that poses and scale bias existing in shape completion methods is clearly stated, which shows the importance of the proposed method.
2. A fully SIM(3)-equivariant network for shape completion is designed to address the identified problem, which includes feature canonicalization to remove translation and scale variation, shape reasoning to eliminate rotation transformations, and transform restoration to recover the absolute alignment. With all of these designs, the proposed network can learn representations agnostic to pose and scale in the training stage, and any transformations to inputs can be applied to the completed outputs.
3. The visualizations in Fig. 5 and Fig.6 show the effectiveness of the proposed network, which can generate completed 3D shapes with more details.


Weaknesses:
1. The training time and datasets used for training are not clearly stated in the paper.
2. It would be better if the authors could show the outputs given the same input partial shape with different poses and scales, which could better illustrate the proposed network is agnostic with pose and scale.
3. I am curious about the performance of the proposed network if the GT completed shape with many thin structures.

---

> ### Author Rebuttal · Authors · 2025-07-28
>
> Dear Reviewer ztk1,
>
> Thank you for your comments. We address your questions below.
>
> > The training time and datasets used for training are not clearly stated in the paper.
>
> Thank you for pointing out this omission. The datasets are specified in Sec. 4.1 under "Datasets and baselines" and include PCN [1], KITTI [2], and OmniObject3D [3]. As detailed in Suppl. Mat. D ("Implementation Details"), all methods were trained for 200 epochs on the PCN dataset using two NVIDIA A40 GPUs. With a batch size of 40, SIMECO's average training time is approximately 63 min/epoch. We have added these details in the revised manuscript.
>
> > It would be better if the authors could show the outputs given the same input partial shape with different poses and scales, which could better illustrate the proposed network is agnostic with pose and scale.
>
> In Fig. 4, we demonstrate SIM(3) equivariance by applying diverse similarity transforms (rotations, translations, and scalings) to the same partial input and showing that the completed outputs follow these transforms exactly. For more extreme pose and scale variations, Fig. 13 in the Suppl. Mat. spans a full page with extensive examples, further confirming that our method consistently maintains high reconstruction quality under large input perturbations. We have also provided an interactive demo in the Suppl. Mat. A (L823) where readers can explore completion results under arbitrary SIM(3) transformations, and we warmly invite you to try it out.
>
> > I am curious about the performance of the proposed network if the GT completed shape with many thin structures.
>
> To evaluate performance on shapes with pronounced thin structures, we compute a local PCA-based anisotropy score:
> $$
> L = \frac{\sigma_1 - \sigma_2}{\sigma_1}
> $$
> on each point's $k$-nearest neighbors ($k=30$), where
> $\sigma_1 \geq \sigma_2 \geq \sigma_3$ are the singular values of the local covariance matrix. A high $L$ indicates a spindly, edge-like neighborhood. We then select shapes in which more than 0.5\% of points satisfy $L > 0.8$, yielding 50 thin-structure cases out of 1200 total. Quantitatively, on this subset, our method achieves an average CD-$\ell1$ of 6.83, compared to 8.59 over the entire test set, demonstrating even better overall performance on such samples with thin structures. Moreover, qualitative examples such as chair and table legs in Fig. 5 and Fig. 6 further confirm that our method preserves fine details effectively.
>
> Thank you again for your insightful feedback. We hope our response has resolved your concerns and are happy to clarify further if needed.
>
> References:
>
> [1] Yuan et al. PCN: Point Completion Network. 3DV 2018.
>
> [2] Geiger et al. Vision Meets Robotics: The KITTI Dataset. IJRR 2013.
>
> [3] Wu et al. OmniObject3D: Large-Vocabulary 3D Object Dataset for Realistic Perception, Reconstruction and Generation. CVPR 2023.

---

> > ### Comment · Reviewer_ztk1 · 2025-08-04
> > **I will maintain my original positive rating**
> >
> > I thank the authors for their comprehensive responses, which have satisfactorily addressed all of my initial comments. I suggest including these discussions in the final version for completeness. I will therefore maintain my original positive rating.

---

> > > ### Author Response · Authors · 2025-08-04
> > > **Thanks for your reply**
> > >
> > > Thank you for your response and for acknowledging our efforts. We’re glad our rebuttal addressed all your comments and will include these discussions in the final version.

---

### Official Review · Reviewer_eyhY · 2025-07-01

**Clarity:** 3
**Significance:** 3
**Originality:** 3
**Rating:** 4
**Confidence:** 5

**Summary:**

This paper aims to address the issues that many point cloud completion methods leak pose and scale cues to the model so that networks may exploit to memorize absolute positions rather than inferring intrinsic geometry. The authors proposed the first shape completion architecture whose modules are SIM(3)-equivariant by design. The experiment shows the good performance of this method.

**Questions:**

1. what is the computation cost of the proposed method

**Ethical Concerns:**

["NO or VERY MINOR ethics concerns only"]

**Final Justification:**

My final justification is maintaining my positive rating.

**Limitations:**

yes

**Quality:**

3

**Strengths And Weaknesses:**

Strengths:
1. The problem setting is very interesting.
2. The formulation is clear and intetersting.

Weaknesses:

1. SIM(3)-equivalence does not convey any meaning if the reader does not know the concept. The author may need to give formal definition, clear introduction earlier or reference for readers to understand what the method aims to do and what was proposed. Similar to SO(3) and SE(3)

2. The compared baseline are out-of-date. The authors need to compare with more recent baselines.

3. Insufficient experiment: For De-biased benchmark evaluation, this paper only provided results on the PCN dataset, which is insufficient as most recent point cloud completion paper will also present results on other ShapeNet-derived datasets to comprehensively evaluate their method.

4. Concern about zero-shot concern: It is invalid to claim it works on zero-shot case. It was tested on KITTI cars, while cars are included in the PCN dataset, therefore it is totally invalid to claim it is zero-shot. Also, many previous methods were also tested on KITTI, but, to my best knowledge, no one claimed that this is zero-shot.

---

> ### Author Rebuttal · Authors · 2025-07-28
>
> Dear Reviewer eyhY,
>
> Thank you for your comments. We address your concerns and questions below.
>
> > SIM(3)-equivalence does not convey any meaning if the reader does not know the concept. The author may need to give formal definition, clear introduction earlier or reference for readers to understand what the method aims to do and what was proposed. Similar to SO(3) and SE(3).
>
> Thank you for this suggestion. In the revised manuscript, we have added an introductory explanation of SIM(3) equivariance, explicitly contrasting it with SO(3) and SE(3), before presenting the formal definition in Sec. 3.1.
>
>
> > The compared baseline are out-of-date. The authors need to compare with more recent baselines.
>
> We focus our comparison on equivariant methods, as they provide the fairest baseline for assessing the impact of equivariance. The most recent equivariant method to our knowledge, ESCAPE [1], supports only SE(3) equivariance and underperforms AdaPoinTr [2] when the latter is trained with data augmentation. In contrast, our method guarantees full SIM(3) equivariance by design and outperforms both ESCAPE (even under its SE(3) setting) and the augmented AdaPoinTr. Since both our method and ESCAPE use the same AdaPoinTr backbone, these comparisons are directly equitable. While more recent non-equivariant methods may show stronger performance under biased settings, they do not address the core issue of pose and scale bias. Our equivariance mechanism is backbone-agnostic and directly remedies these biases, making our method a principled solution to fundamental limitations in existing shape completion pipelines, rather than merely another state-of-the-art variant.
>
> Nevertheless, to provide additional context, we also evaluated ODGNet [3], a recent non-equivariant method. As shown in the table below, under the same debiased setting, our method consistently outperforms ODGNet across all categories and metrics, further highlighting the advantage of built-in SIM(3)-equivariance.
>
> |Method|Transform|Airpl.|Cab.|Car|Chair|Lamp|Sofa|Table|Wat.|CD-$\ell_1$↓|F1↑|
> |---|---|---|---|---|---|---|---|---|---|---|---|
> |AdaPoinTr [2]|I/I|3.68|8.82|7.47|6.85|5.47|8.35|5.80|5.76|6.53|0.845|
> |ODGNet [3]|I/I|3.77|8.77|7.56|6.84|5.09|8.47|5.84|5.66|6.50|0.833|
> |AdaPoinTr [2]|I/SIM(3)|31.93|78.90|63.51|60.56|61.47|70.77|71.94|42.74|60.23|0.206|
> |ODGNet [3]|I/SIM(3)|15.03|52.36|41.04|39.86|30.22|50.01|46.73|24.45|37.46|0.296|
> |AdaPoinTr [2]|SIM(3)/SIM(3)|6.46|12.17|10.51|10.29|7.59|12.26|8.90|8.14|9.54|0.661|
> |ODGNet [3]|SIM(3)/SIM(3)|6.16|11.60|11.15|10.13|6.81|13.12|9.48|8.19|9.58|0.659|
> |**SIMECO (ours)**|I/SIM(3)|**6.02**|**10.75**|**9.27**|**9.25**|**6.66**|**11.16**|**7.82**|**7.77**|**8.59**|**0.714**|
>
> > Insufficient experiment: For De-biased benchmark evaluation, this paper only provided results on the PCN dataset, which is insufficient as most recent point cloud completion paper will also present results on other ShapeNet-derived datasets to comprehensively evaluate their method.
>
> We agree that evaluating on other ShapeNet-derived benchmarks can be helpful. However, our primary goal is to identify and understand the theoretical benefits of equivariance itself, rather than to chase state-of-the-art scores across all datasets. To this end, we matched exactly the three benchmarks used by the recent method ESCAPE [1] to enable a direct, apples-to-apples comparison, but also focused our compute budget on a detailed analysis of when and why equivariance delivers gains (Tabs. 4-5, Figs. 8-9). We believe this targeted evaluation on PCN [4], KITTI [5], and OmniObject3D [6], together with dense ablations and analyses, provides meaningful insight into the fundamental behavior of equivariance in shape completion.
>
>
> > Concern about zero-shot concern: It is invalid to claim it works on zero-shot case. It was tested on KITTI cars, while cars are included in the PCN dataset, therefore it is totally invalid to claim it is zero-shot. Also, many previous methods were also tested on KITTI, but, to my best knowledge, no one claimed that this is zero-shot.
>
> Thank you for pointing this out. You are correct that "zero-shot" may suggest unintended implications. To align with prior works, we have removed all instances of "zero-shot" throughout the manuscript.
>
> > What is the computation cost of the proposed method?
>
> We report computational efficiency in Suppl. Mat. B.2, with average per-scan latencies in Table 7. Despite offering full SIM(3)-equivariance, our method processes a scan end-to-end in just 76 ms—roughly half the runtime of the next-fastest equivariant method, ESCAPE [1] (148 ms), and even faster than that of EquivPCN [7] and SCARP [8] (172 ms each). ESCAPE and SCARP spend extra time in post-processing steps, which inflate total latency. Although AdaPoinTr runs faster at 16 ms, it does so without any built-in equivariance. Overall, our method achieves the best balance of speed and rigorous SIM(3) symmetry preservation.
>
> Thank you again for your insightful feedback. We hope our response has resolved your concerns and are happy to clarify further if needed.
>
> References:
>
> [1] Bekci et al. ESCAPE: Equivariant Shape Completion via Anchor Point Encoding. CVPR 2025.
>
> [2] Yu et al. AdaPoinTr: Diverse Point Cloud Completion with Adaptive Geometry-aware Transformers. TPAMI 2023.
>
> [3] Cai et al. Orthogonal Dictionary Guided Shape Completion Network for Point Cloud. AAAI 2024.
>
> [4] Yuan et al. PCN: Point Completion Network. 3DV 2018.
>
> [5] Geiger et al. Vision Meets Robotics: The KITTI Dataset. IJRR 2013.
>
> [6] Wu et al. OmniObject3D: Large-Vocabulary 3D Object Dataset for Realistic Perception, Reconstruction and Generation. CVPR 2023.
>
> [7] Wu and Miao. SO(3) Rotation Equivariant Point Cloud Completion Using Attention-Based Vector Neurons. 3DV 2022.
>
> [8] Sen et al. SCARP: 3D Shape Completion in Arbitrary Poses for Improved Grasping. ICRA 2023.

---

> > ### Comment · Reviewer_eyhY · 2025-08-04
> >
> > I appreciate the authors' efforts in providing additional results and detailed explanations. The response addressed my concerns, and I will maintain my positive rating.

---

> > > ### Author Response · Authors · 2025-08-04
> > > **Thanks for your reply**
> > >
> > > Thank you for acknowledging our efforts and maintaining a positive rating. We’re pleased our rebuttal addressed your concerns.

---

### Official Review · Reviewer_AFMZ · 2025-07-02

**Clarity:** 3
**Significance:** 3
**Originality:** 3
**Rating:** 5
**Confidence:** 4

**Summary:**

This paper claims that they developed a SIM(3)-equivariant point cloud completion network. It aims to address the problem of shape completion in any reference frame, which is a very important problem, in my opinion. The method is through translation and scale normalization, building on top of the Vector Neurons feature representation. Experiments are conducted under a "debiased protocol", meaning that the bounding box and center information are not given during inference, which aligns much better with the practical use case.

**Questions:**

Explain how the vector features are initialized for each point, and discuss the equivariance property at this step.

The layer normalization seems to be done per point, but why is the scale restoration common across all points? Does that create a discrepancy?

Clarify the dimensions in the canonicalization step.

Explain equation 47 in the appendix. Why is it designed this way? It seems to be different from typical VN-DGCNNs. What is the dimension of $\bar{V}_i^l$, and what's its meaning? The per-point mean across the feature dimension?

Is the definition of $mu^l$ wrong? The $E_i$ should be outside of the norm? Is $E$ expectation?

**Ethical Concerns:**

["NO or VERY MINOR ethics concerns only"]

**Final Justification:**

The author addressed my concerns. I recommend accepting this paper.

**Limitations:**

yes

**Quality:**

3

**Strengths And Weaknesses:**

I'm glad that this paper aimed to address the issue of shape completion from arbitrary reference frames without privileged ground-truth information in terms of the center or scale. This is the real challenge in practice, and solving it adds real value.

In 3D shape completion, I think we need to be careful about what equivariance we are talking about. The challenge of applying equivariance in 3D point clouds mainly comes from the partial observation. It means that the observations from different perspectives are no longer simply connected by a transformation, and it also brings ambiguities in terms of the reference frame. In this context, discussing equivariance to the transformation of a single input partial-view point cloud itself is not very useful, because you can always centralize and scale-normalize the point cloud, leaving only rotation variations to be concerned with, in which case, only rotation equivariance is needed. However, what's more important is finding the internal symmetries among different partial observations. While they can be associated using a common center, the underlying center is not in the data, and centering each input individually breaks the association across observations. Furthermore, the scale-normalization step, which is common in point cloud learning, leads to further discrepancy across observations, because different parts have different scales. The above calls for translation and scaling equivariance for the shape completion task.

I don't think any work, including this work, has a perfect solution to the equivariance discussed above, because the centering and scaling information are not directly observable in a single input point cloud. However, most previous work (cited in this paper) only attempted to approximate translation equivariance (for example, using anchor points as in ESCAPE [10]), but left scaling out. This paper does not do any better in dealing with translation equivariance (it still relies on centering the input point cloud), but the scale normalization in the layer-normalization step, done for the features of each point, addressed the scaling variations in the partial view point clouds after the unit-ball normalization. I think this is the key contribution and differentiator of this work.

The proposed strategy showed its practical value in the experiments under the realistic evaluation protocol, which I appreciate.

Weaknesses:
As in the previous discussion, this paper also followed the notation of equivariance of a single point cloud, and thus, the proof of equivariance is equally trivial and impractical as previous work. Specifically, the problem still lies in the translation. For different partial observations, the centering step subtracts a different offset, resulting in different outcomes, thus breaking the symmetry.

You may argue that the per-point feature canonicalization removed the dependency of perfect centralization. However, if I understand correctly, the vector features for each point in Vector Neurons are initialized with the absolute coordinate of each point and the local differences with other points. Global translation will change the global component but not the local component, resulting in different overall vector features, which cannot be cancelled by point-wise canonicalization.

The notations in the method explanation are unclear, especially in terms of dimensions. For example, the canonicalization step is written with a lot of ambiguities. What are "channel-wise mean", "row-wise norm", and "modified layer normalization"? I need to think very carefully to make sense of these descriptions. Use letters to disambiguate the dimensions. Figure 3 also didn't help too much.

---

> ### Author Rebuttal · Authors · 2025-07-28
>
> Dear Reviewer AFMZ,
>
> Thank you for your comments. We address your concerns and questions below.
>
> Throughout the discussion, we denote shape features as $\mathcal{V} = \\{\mathbf{V}\_i\\}\_{i=1}^M$, where $\mathbf{V}\_i \in \mathbb{R}^{D \times 3}$. Here, $M$ denotes the number of vector features, $D$ the feature dimension, and $3$ the vector dimension.
>
> > Specifically, the problem still lies in the translation. For different partial observations, the centering step subtracts a different offset, resulting in different outcomes, thus breaking the symmetry. You may argue that the per-point feature canonicalization removed the dependency of perfect centralization. However, if I understand correctly, the vector features for each point in Vector Neurons are initialized with the absolute coordinate of each point and the local differences with other points. Global translation will change the global component but not the local component, resulting in different overall vector features, which cannot be cancelled by point-wise canonicalization.
>
> > This paper does not do any better in dealing with translation equivariance (it still relies on centering the input point cloud)
>
> We believe this impression may stem from a misunderstanding of our method design. Unlike earlier vector neuron-based methods [1, 2, 3] which **explicitly** subtract the input centroid and thus result in reference frame ambiguities, our method does not rely on any centering of the input points' coordinates. Instead, we follow the practice of VN-SPD [4] by constraining $\mathbf{W} \mathbf{1}\_D =\mathbf{1}\_{D'}$, where  $\mathbf{W} \in \mathbb{R}^{D'\times D}$ is the weight matrix for linear layers, and design each SIM(3)-equivariant block to preserve strict equivariance. This way, no ambiguity exists because the network can **inherently** distinguish changes in input arising from translation or from intrinsic geometry. For a full description of our SIM(3)-equivariant architecture, please see Suppl. Mat. C.2 (Propositions 1-3).
>
> We then further clarify below two pieces of detail that might have caused the misunderstanding.
>
> > Explain how the vector features are initialized for each point, and discuss the equivariance property at this step.
>
> Each point's vector features are initialized as in Eq. (47):
>
> $$
>  \mathrm{VN\text{-}Max}\_{j \in \mathcal{N}\_i} \left( \mathrm{VNLA}
> \left((\mathbf{x}\_j-\mathbf{x}\_i+\bar{\mathbf{x}}\_i)\oplus \mathbf{x}\_i \right)\right),
> $$
>
> where $\mathcal{N}\_i$ is the $k$-nearest neighbors of point $i$, and $\oplus$ denotes feature concatenation. $\mathrm{VNLA}(\cdot)$ applies $\mathrm{VN\text{-}Linear}(\cdot)$ followed by $\mathrm{VN\text{-}LeakyReLU}(\cdot)$. $\bar{\mathbf{x}}\_i \in \mathbb{R}^{1\times 3}$ is the mean coordinate of all input points  $ \mathbf{x} =\\{x\_i \in \mathbb{R}^3\\}\_{i=1}^{N\_\mathrm{in}}$,
> which preserves translation offset for the local difference $(\mathbf{x}\_j-\mathbf{x}\_i+ \bar{\mathbf{x}}\_i)$.
>
> For all $g \in \mathrm{SIM}(3)$,
> $$
> g \cdot \left(( \mathbf{x}_j-\mathbf{x}_i + \bar{\mathbf{x}}_i) \oplus \mathbf{x}_i \right)
> = \left( ((s R \mathbf{x}_j+t)-(s R\mathbf{x}_i+t)+(s R \bar{\mathbf{x}}_i+t))\oplus (sR\mathbf{x}_i +t) \right) \notag
> = sR\left((\mathbf{x}_j-\mathbf{x}_i+\bar{\mathbf{x}}_i )\oplus \mathbf{x}_i \right)+t
> $$
>
> Thus, the initialized vector features remain SIM(3)-equivariant.
>
> > Explain equation 47 in the appendix. Why is it designed this way? It seems to be different from typical VN-DGCNNs. What is the dimension of $\bar{\mathbf{V}}\_i^l$, and what's its meaning? The per-point mean across the feature dimension?
>
> As you correctly noted, typical VN-DGCNN initializes vector features with $\left( (\mathbf{x}\_j-\mathbf{x}\_i) \oplus \mathbf{x}\_i \right)$ and constructs vector features per layer in the same way $\left( (\mathbf{V}\_j^l-\mathbf{V}\_i^l) \oplus \mathbf{V}\_i^l \right)$. While $\mathbf{V}\_i^l $ is translation-equivariant, the subtraction $(\mathbf{V}\_j^l-\mathbf{V}\_i^l)$ cancels the translation term $t$, which would break the equivariance. To address this, we follow the practice from VN-SPD [4] (Eq. (47)),
> $$\mathrm{VN\text{-}Max}_{j \in \mathcal{N}_i} \left( \mathrm{VNLA} \left( (\mathbf{V}_j^l - \mathbf{V}_i^l+\bar{\mathbf{V}}_i^l) \oplus \mathbf{V}_i^l \right) \right),$$
> where  $\bar{\mathbf{V}}\_i^l \in \mathbb{R}^{D^l \times 3}$ is the mean across $M$ features and is added back to the local difference $(\mathbf{V}\_j^l-\mathbf{V}\_i^l+\bar{\mathbf{V}}\_i^l)$ to restore $t$.
>
> > As in the previous discussion, this paper also followed the notation of equivariance of a single point cloud, and thus, the proof of equivariance is equally trivial and impractical as previous work.
>
> We appreciate the opportunity to clarify our perspective. As previously explained, our strict SIM(3) equivariance ensures that we never need to centralize input points as prior works typically do. For the same object, all partial observations, their learned equivariant features, and the outputs, naturally reside in the original object frame. Importantly, any higher-level symmetry across arbitrary observations cannot be enforced explicitly and must instead be learned, as equivariance can only be defined per observation and can only help us this far. Our approach embraces this principle by letting the network learn truly intrinsic geometric features that are inherently invariant to pose and scale, rather than forcing all inputs into a common reference frame.
>
> > The notations in the method explanation are unclear, especially in terms of dimensions. For example, the canonicalization step is written with a lot of ambiguities. What are "channel-wise mean", "row-wise norm", and "modified layer normalization"? Use letters to disambiguate the dimensions.
>
> > Clarify the dimensions in the canonicalization step.
>
> Thank you for pointing out the ambiguity and your suggestion.
>
> The "channel-wise mean" $\bar{\mathbf{V}}\_{i} $ is defined as
>
> $$
> \bar{\mathbf{V}}\_{i} = \frac{1}{D} \sum\_{d=1}^{D} \mathbf{V}\_{i,d},\quad\bar{\mathbf{V}}\_{i}\in\mathbb{R}^{1\times3}.
> $$
>
> The "row-wise norm" $
> \big\Vert\mathbf{V}\_{i}-\bar{\mathbf{V}}\_{i}\big\Vert\_{2}
> $  is defined as the $\ell\_2$-norm across 3 vector dimensions:
>
> $$
> \Vert\mathbf{V}\_{i}-\bar{\mathbf{V}}\_{i}\big\Vert\_{2} = \sqrt{\sum\_{k=1}^{3} \left( \mathbf{V}\_{i}-\bar{\mathbf{V}}\_{i} \right)^{2}\_{:,k}}
> , \quad
> \big\Vert\mathbf{V}\_{i}-\bar{\mathbf{V}}\_{i}\big\Vert\_{2} \in \mathbb{R}^{D \times 1}.
> $$
>
> The "modified layer normalization" corresponds to Eq. (6), ${\mathbf{V}^{\prime}}\_{i}
> = \mathrm{layernorm}(\big\Vert\mathbf{V}\_{i} -\bar{\mathbf{V}}\_{i}\big\Vert_2)  \frac{\mathbf{V}\_{i}-\bar{\mathbf{V}}\_{i}}{\big\Vert\mathbf{V}\_{i}-\bar{\mathbf{V}}\_{i}\big\Vert\_2}$,
> where the $\mathrm{layernorm}$ operation is applied to the scalar-valued "row-wise norm"
> $\big\Vert\mathbf{V}\_{i}-\bar{\mathbf{V}}\_{i}\big\Vert\_{2}$. $\big\Vert\mathbf{V}\_{i} - \bar{\mathbf{V}}\_{i}\big\Vert\_{2}$ produces scalar neurons for standard layer normalization. We have revised the manuscript to explicitly use letters for solving the ambiguity over dimensions.
>
> > The layer normalization seems to be done per point, but why is the scale restoration common across all points? Does that create a discrepancy?
>
> The layer normalization (canonicalization) serves two purposes:
> (1) For the same partial scan under SIM(3) transformation, the canonicalization step can factor out the global translation and scale from the shape features (Proposition 4 in the Suppl. Mat.):
> $
> \mathrm{VN\text{-}LayerNorm}(s \mathbf{V}R + t)=\mathrm{VN\text{-}LayerNorm}(\mathbf{V}) R.$
> (2) It performs the mean-centering $(\mathbf{V}-\bar{\mathbf{V}})$ and variance normalization of standard layer normalization, extended to vector features, which stabilizes the attention mechanism during training.
>
> The scale restoration is used to align the completion result with the original sensor frame, that is, to recover $s$. We apply this single, common scale only after the attention-based shape reasoning, since the relative magnitudes of feature vectors encode intrinsic geometric information. Introducing per-feature scaling would re-entangle the intrinsic geometry with the external transform and potentially disrupt the attention. By using one global scale, we preserve the relative magnitudes while correctly restoring the output to the sensor frame.
>
> > Is the definition of $\mu\_l$ wrong? The $E\_i$ should be outside of the norm? Is $E$ expectation?
>
> $E$ is indeed empirical expectation. The definition of $\mu\_l$ is correct as we intentionally place the point-wise expectation $E\_i$ inside the norm.
> $E\_i(\mathbf{V}^{l}\_{i}-\bar{\mathbf{V}}^{l}\_{i}) \in \mathbb{R}^{D \times 3}$ is the  mean across $M$ features and $E\_{D^l}$ is the mean across $D$ dimensions for $\big\Vert E\_i(\mathbf{V}^{l}\_{i}-\bar{\mathbf{V}}^{l}\_{i})\big\Vert\_2 \in \mathbb{R}^{D}$. Taking the mean across $M$ features first provides a stable global representation, and applying the $\ell\_2$-norm across the vector dimension before averaging over channels ensures that $\mu\_l$ purely reflects the magnitude and ensures stability. We have revised the manuscript to use $\mathbb{E}[\cdot]$ more consistently.
>
> Thank you again for your insightful feedback. We hope our response has resolved your concerns and are happy to clarify further if needed.
>
> References:
>
> [1] Lei et al. EFEM: Equivariant Neural Field Expectation Maximization for 3D Object Segmentation without Scene Supervision. CVPR 2023.
>
> [2] Yang et al. EquiBot: SIM(3)-Equivariant Diffusion Policy for Generalizable and Data Efficient Learning. CoRL 2024.
>
> [3] Yang et al. EquivAct: SIM(3)-Equivariant Visuomotor Policies beyond Rigid Object Manipulation. ICRA 2024.
>
> [4] Katzir et al. Shape-Pose Disentanglement using SE(3)-Equivariant Vector Neurons. ECCV 2022.

---

> > ### Comment · Reviewer_AFMZ · 2025-08-03
> >
> > Thanks for the clarification, which helped me understand the method more clearly. However, they didn't fully resolve my concerns.
> >
> > First, the fact that the feature initialization relies on the global point cloud mean implies that it will encounter feature inconsistency for different partial observations of the same object, due to the different global point cloud means even without explicit translation, which is the same as previous methods that conducts global average subtraction at the beginning. It means that the equivariance property only holds for the exact same partial observation up to SIM(3) transformations.
> >
> > Second, the restoration step is still confusing to me. Taking average of $M$ point features in their vector form will likely cancel out most features since the average is over centralized point features. It makes more sense to me if the vector mean is over the D channels and the norm mean is over the M points, because the D channels likely won't cancel each other. Could you explain why you made your choice?

---

> > > ### Author Response · Authors · 2025-08-04
> > > **Response to two remaining points**
> > >
> > > Thank you for your reply. Your feedback brings us closer to a shared understanding, and we’re happy to address your remaining concerns:
> > >
> > > **First**, we'd like to further clarify the fundamental difference between input centering and our design with strict translation equivariance when it comes to different partial observations of the same object, with an example.
> > >
> > > Let $\mathbf{x}^a$ and $\mathbf{x}^b$ be two partial observations of the same object with global means $\bar{\mathbf{x}}^a$ and $\bar{\mathbf{x}}^b$, and let $p\in\mathbf{x}^a\cap\mathbf{x}^b$ be a shared point. We denote point-wise features for brevity.
> > > Previous methods apply a centering at the beginning:
> > >
> > > $$
> > > f\_{\mathrm{prev}}(p\mid\mathbf{x}^a) = f\_{\mathrm{prev}}(p - \bar{\mathbf{x}}^a\bigr),
> > > \quad
> > > f\_{\mathrm{prev}}(p\mid\mathbf{x}^b) = f\_{\mathrm{prev}}(p - \bar{\mathbf{x}}^b\bigr)
> > > $$
> > >
> > > $$\Longrightarrow\quad
> > > f\_{\mathrm{prev}}(p\mid\mathbf{x}^a)\neq f\_{\mathrm{prev}}(p\mid\mathbf{x}^b),
> > > $$
> > >
> > > so even the same point $p$ is assigned two translation-entangled, inconsistent features with no way to recover a consistent representation.
> > >
> > > By contrast, our method disentangles intrinsic and translation components:
> > >
> > > $$
> > > f\_{\mathrm{ours}}(p\mid\mathbf{x}^a) = \bigl[f\_{\mathrm{ours}}^{\ast}(p)\,T(p\mid\bar{\mathbf{x}}^a)\bigr],
> > > \quad
> > > f\_{\mathrm{ours}}(p\mid\mathbf{x}^b) = \bigl[f\_{\mathrm{ours}}^{\ast}(p)\,T(p\mid\bar{\mathbf{x}}^b)\bigr],
> > > $$
> > >
> > > $$
> > > \Longrightarrow\quad
> > > f\_{\mathrm{ours}}^{\ast}(p\mid\mathbf{x}^a) = f\_{\mathrm{ours}}^{\ast}(p\mid\mathbf{x}^b).
> > > $$
> > >
> > > The intrinsic feature $f\_{\mathrm{ours}}^{\ast}(p)$ is fully translation-invariant and remains identical across the observations. Any change in the global mean $\bar{\mathbf{x}}$ affects only the separate translation term $ T(p \mid \bar{\mathbf{x}}) $ and never alters the intrinsic representation. Consequently, different offsets from initialization never introduce inconsistency into $f\_{\mathrm{ours}}^{\ast}(p)$ during learning.
> > >
> > > To summarize, previous centering-based methods map the same point $p$ to different features whenever $\bar{\mathbf{x}}$ differs and thus create ambiguity. In contrast, our method decomposes each feature into an intrinsic component and a separate translation component. Because $f\_{\mathrm{ours}}^{\ast}(p)$ is invariant to any global shift, a shared point $p$ yields identical intrinsic features across different observations. All shape reasoning relies exclusively on these intrinsic features. **Even though initialization depends on $\bar{\mathbf{x}}$, the intrinsic representation remains fully agnostic to $\bar{\mathbf{x}}$.** This ability to maintain consistent intrinsic representations across observations is a core advantage of our approach and completely eliminates the ambiguity inherent in simple centering.
> > >
> > > ---
> > >
> > > **Second**, taking the average of the point features in their vector form doesn't cancel out features, as it is a standard global mean pooling operation to aggregate all point-wise features into a single global embedding. Scale is an object-level attribute and must be restored from the global feature embedding, not from point-wise feature magnitudes. For example, the size of a single tire does not necessarily reveal the scale of the whole car. Normalizing each feature first would detach it from the vector shape feature space, and the subsequent mean pooling would no longer produce a meaningful global embedding. Other VN‐based methods [1, 2, 3] also follow this order of operations to derive scale from a global embedding rather than point-wise magnitudes.
> > >
> > > We note that reusing the notation $\bar{\mathbf{V}}\_{i}^{l}$ as in VN‐DGCNN may have caused the confusion. In $\mu^l = E\_{D^l}
> > > \big\Vert E\_i(\mathbf{V}^{l}\_{i} -\bar{\mathbf{V}}^{l}\_{i})\big\Vert \_2$,   $\bar{\mathbf{V}}^{l}\_{i}$ refers to the
> > > **channel-wise mean** of the feature vector across the $D$ dimensions:
> > > $$
> > > \bar{\mathbf{V}}^{l}\_{i} = \frac{1}{D} \sum\_{d=1}^{D} \mathbf{V}^{l}\_{i,d},
> > > \quad \bar{\mathbf{V}}^{l}\_{i} \in \mathbb{R}^{1 \times 3}.
> > > $$
> > > Hence, if we take the vector mean over the D channels first, $\frac{1}{D} \sum\_{d=1}^{D}(\mathbf{V}^{l}\_{i} -\bar{\mathbf{V}}^{l}\_{i}) =\bar{\mathbf{V}}^{l}\_{i}  - \bar{\mathbf{V}}^{l}\_{i} = 0 $. We have revised the manuscript to avoid this ambiguity.
> > >
> > > ---
> > >
> > > We’re here for the rest of the discussion period and would be happy to clarify anything further!

---

> > > > ### Comment · Reviewer_AFMZ · 2025-08-05
> > > >
> > > > For the first point, what is $f_{\mathrm{ours}}^{\ast}(p)$ and $T(p\mid\bar{\mathbf{x}}^a)$ exactly in your method? Do they have correspondence in (47) in the feature initialization stage?
> > > >
> > > > For the second point, now I understand. The confusion indeed comes from the reuse of the $\bar{V}^l_i$ notation.

---

> > > > > ### Author Response · Authors · 2025-08-05
> > > > > **Response to intrinsic feature consistency**
> > > > >
> > > > > $f\_{\mathrm{ours}}^{\ast}(p)$ denotes the final intrinsic shape feature of point \(p\), specifically the attention output $\mathbf{Z}^{L\_e}$ (L170) from the last encoder layer ($L\_e$). $T(p\mid\bar{\mathbf{x}}^a)$ denotes the translation component from the same layer, specifically $\mathbf{V}^{L\_e}$ from Eq. (9). Both are final encoder representations and do not directly correspond to Eq. (47) which addresses only initialization.
> > > > >
> > > > > Our VN-DGCNN design, including initialization, follows the VN-SPD practice [4] and achieves strict shape-pose disentanglement only when considered as a complete, end-to-end network. As a result,  by construction, the final intrinsic representation $f_{\mathrm{ours}}^{\ast}(p)$ remains fully agnostic to $\bar{\mathbf{x}}$ (translation). Although the initialization from the two observations differs, it does not affect the **final** intrinsic feature. Intuitively, the same physical point in both observations yields the same intrinsic representation, depending solely on the local (relative) geometry. For full details on the disentanglement mechanism, which differs from simple centering, please refer to VN-SPD [4].
> > > > >
> > > > > ---
> > > > >
> > > > > We hope this clarifies the remaining point and are happy to address any further questions!

---

> > > > > > ### Author Response · Authors · 2025-08-06
> > > > > > **Response to intrinsic feature consistency (continued)**
> > > > > >
> > > > > > For your convenience we quote two passages from VN-SPD [4]:
> > > > > > >  While translation invariance could theoretically be achieved by centering the input point clouds, such approach is sensitive to noise, missing data and partial shapes. Therefore we propose an extension to VNN achieving SE(3)-equivariance.
> > > > > >
> > > > > > > Equivariance to translation, on the other-hand, allows identifying local features in different locations with the same filters, without requiring global parameters.
> > > > > >
> > > > > > In summary, we believe this form of translation equivariance differs fundamentally from simple centering and avoids the resulting inconsistencies at the level of intrinsic point-wise features. Nevertheless, we acknowledge that assuming fully point-wise features is a simplification, and achieving strictly consistent features across observations remains a challenge. We'd be delighted to hear any further thoughts you may have!

---

> > > > > > > ### Comment · Reviewer_AFMZ · 2025-08-08
> > > > > > >
> > > > > > > Thank you for the further explanation, but I still don't fully agree with your claim.
> > > > > > >
> > > > > > > The VN-SPD paper was not specifically designed to handle partial observations, so the translation equivariance under occlusion cannot be directly explained by citing that paper. Specifically, as you said in the initial rebuttal when explaining (47), $\bar{x}_i\in \mathbb{R}^{1\times 3}$ is the mean coordinate of all input points. This coordinate changes across different partial observations of the same object at the same pose, resulting in non-equivariant feature initialization. This is similar to the issue that the centering-based VNN methods face.

---

> > > > > > > > ### Author Response · Authors · 2025-08-08
> > > > > > > > **Further clarification with supporting ablation results**
> > > > > > > >
> > > > > > > > Thank you for the follow-up. We agree that feature initialization can vary across different partial observations and that, without strong assumptions, this issue is not fully resolved. Our claim was intended to clarify two points:
> > > > > > > >
> > > > > > > > * *No input centering*. We do not recenter inputs as previous methods do.
> > > > > > > >
> > > > > > > > * *Principled optimization*. With VN-SPD we achieve a more principled optimization than centering-based VNN. To substantiate this claim, in addition to our previous theoretical discussion, we performed an ablation in which we replaced the VN-SPD constraint with standard VN layers combined with centering and normalization on partial inputs (i.e., same as a centering-based VNN). As shown below, our full model significantly outperforms the centering variant.
> > > > > > > >
> > > > > > > > | Method                     | Airpl. | Cab.  | Car   | Chair  | Lamp  | Sofa  | Table  | Wat.   | CD-ℓ₁ ↓ | F1 ↑  |
> > > > > > > > |----------------------------|--------|-------|-------|--------|-------|-------|--------|--------|----------|-------|
> > > > > > > > | AdaPoinTr (augmented)      | 6.46   | 12.17 | 10.51 | 10.29  | 7.59  | 12.26 | 8.90   | 8.14   | 9.54     | 0.661 |
> > > > > > > > | SIMECO (centered)          | 6.10   | 11.77 | 10.24 | 10.05  | 7.50  | 12.16 | 8.57   | 8.10   | 9.31     | 0.673 |
> > > > > > > > | **SIMECO (ours)**          | **6.02** | **10.75** | **9.27** | **9.25** | **6.66** | **11.16** | **7.82** | **7.77** | **8.59**  | **0.714** |
> > > > > > > >
> > > > > > > > We fully acknowledge that both approaches face *similar* limitations as neither fully resolves initialization variability as you noted, but they still differ in mechanism and empirical performance. This is why we opted for VN-SPD rather than simple centering.
> > > > > > > >
> > > > > > > > We appreciate this in-depth discussion and hope these clarifications help us reach a consensus. As always, we welcome any further feedback you may have!

---

> > > > > > > > > ### Comment · Reviewer_AFMZ · 2025-08-08
> > > > > > > > >
> > > > > > > > > I agree that the VN-SPD approach should bring improvements over centering, as the origin can be learned instead of fixed, which can partly compensate for the limitation we are discussing here. I also appreciate the added experimental result.
> > > > > > > > >
> > > > > > > > > I wanted to make this point clear because the equivariance claim in the original paper was overclaimed, especially in the context of partial geometry processing, given what we agreed upon now. I would love to see this caveat mentioned in the final paper. But otherwise, nice paper with strong results on an important problem.

---

> > > > > > > > > > ### Author Response · Authors · 2025-08-09
> > > > > > > > > > **Thanks for the constructive discussion**
> > > > > > > > > >
> > > > > > > > > > Thank you for the constructive discussion and for recognizing the merits of our work. We will clearly state the agreed caveat in the final paper.

---

### Decision · Program_Chairs · 2025-09-17

**Decision:**

Accept (poster)

**Comment:**

The paper proposes a shape completion method which is equivariant to similarity transforms.
All reviewers are in favor of acceptance and note the paper is sound and well evaluated. The AC panel recommends acceptance.